# LABEL ENCODING FOR REGRESSION NETWORKS

**Deval Shah, Zi Yu Xue & Tor M. Aamodt**
Department of Electrical and Computer Engineering
University of British Columbia, Vancouver, BC, Canada
`{devalshah,fzyxue,aamodt}@ece.ubc.ca`

## ABSTRACT

Deep neural networks are used for a wide range of regression problems. However, there exists a significant gap in accuracy between specialized approaches and generic direct regression in which a network is trained by minimizing the squared or absolute error of output labels. Prior work has shown that solving a regression problem with a set of binary classifiers can improve accuracy by utilizing well-studied binary classification algorithms. We introduce *binary-encoded labels (BEL)*, which generalizes the application of binary classification to regression by providing a framework for considering arbitrary multi-bit values when encoding target values. We identify desirable properties of suitable encoding and decoding functions used for the conversion between real-valued and binary-encoded labels based on theoretical and empirical study. These properties highlight a tradeoff between classification error probability and error-correction capabilities of label encodings. BEL can be combined with off-the-shelf task-specific feature extractors and trained end-to-end. We propose a series of sample encoding, decoding, and training loss functions for BEL and demonstrate they result in lower error than direct regression and specialized approaches while being suitable for a diverse set of regression problems, network architectures, and evaluation metrics. BEL achieves state-of-the-art accuracies for several regression benchmarks. Code is available at `https://github.com/ubc-aamodt-group/BEL_regression`.

## 1 INTRODUCTION

Deep regression networks, in which a continuous output is predicted for a given input, are traditionally trained by minimizing squared/absolute error of output labels, which we refer to as *direct regression*. However, there is a significant gap in accuracy between direct regression and recent task-specialized approaches for regression problems including head pose estimation, age estimation, and facial landmark estimation. Given the increasing importance of deep regression networks, developing generic approaches to improving their accuracy is desirable.

A regression problem can be posed as a set of binary classification problems. A similar approach has been applied to other domains such as ordinal regression (Li & Lin, 2006) and multiclass classification (Dietterich & Bakiri, 1995). Such a formulation allows the use of well-studied binary classification approaches. Further, new generalization bounds for ordinal regression or multiclass classification can be derived from the known generalization bounds of binary classification. This reduces the efforts for design, implementation, and theoretical analysis significantly (Li & Lin, 2006). Dietterich & Bakiri (1995) demonstrated that posing multiclass classification as a set of binary classification problems can increase error tolerance and improve accuracy. However, the proposed approaches for multiclass classification do not apply to regression due to the differences in task objective and properties of the classifiers' error probability distribution (Section 2). On the other hand, prior works on ordinal regression have explored the application of binary classifiers in a more restricted way which limits its application to a wide range of complex regression problems (Section 2). *There exists a lack of a generic framework that unifies possible formulations for using binary classification to solve regression.*

In this work, we propose *binary-encoded labels (BEL)* which improves accuracy by generalizing application of binary classification to regression. In BEL, a target label is quantized and converted to a binary code of length $M$, and $M$ binary classifiers are then used to learn these binary-encoded

labels. An encoding function is introduced to convert the target label to a binary code, and a decoding function is introduced to decode the output of binary classifiers to a real-valued prediction. BEL allows using an adjustable number of binary classifiers depending upon the quantization, encoding, and decoding functions. BEL opens possible avenues to improve the accuracy of regression problems with a large design space spanning quantization, encoding, decoding, and loss functions.

We focus on the encoding and decoding functions and theoretically study the relations between the absolute error of label and binary classifiers' errors for sample encoding and decoding functions. This analysis demonstrates the impact of binary classifiers' error distribution over the numeric range of target labels on the suitability of different encoding and decoding functions. Based on our analysis and empirically observed binary classifiers' error distribution, we propose properties of suitable encoding functions for regression and explore various encoding functions on a wide range of tasks. We also propose an expected correlation-based decoding function for regression that can effectively reduce the quantization error introduced by the use of classification.

A deep regression network consists of a feature extractor and a regressor and is trained end-to-end. A regressor is typically the last fully connected layer with one output logit for direct regression. Our proposed regression approach (BEL) can be combined with off-the-shelf task-specific feature extractors by increasing the regressor's output logits. Further, we find that the correlation between multiple binary classifiers' outputs can be exploited to reduce the size of the feature vector and consequently reduce the number of parameters in the regressor. We explore the use of different decoding functions for training loss formulation and evaluate binary cross-entropy, cross-entropy, and squared/absolute error loss functions for BEL. We evaluate BEL on four complex regression problems: head pose estimation, facial landmark detection, age estimation, and end-to-end autonomous driving. We make the following contributions in this work:

- We propose binary-encoded labels for regression and introduce a general framework and a taxonomy for the design aspects of regression by binary classification. We propose desirable properties of encoding and decoding functions suitable for regression problems.

- We present a series of suitable encoding, decoding, and loss functions for regression with BEL. We present an end-to-end learning approach and regression layer architecture for BEL. We combine BEL with task-specific feature extractors for four tasks and evaluate multiple encoding, decoding, and loss functions. BEL outperforms direct regression for all the problems and specialized approaches for several tasks.

- We theoretically and empirically demonstrate the effect of different design parameters on the accuracy, how it varies across different tasks, datasets, and network architectures, and provide preliminary insights and motivation for further study.

## 2  RELATED WORK

**Binary classification for regression:**  Prior works have proposed binary classification-based approaches for ordinal regression  (Crammer & Singer, 2001; Chu & Keerthi, 2005; Li & Lin, 2006). Ordinal regression is a class of supervised learning problems, where the samples are labeled by a rank that belongs to an ordinal scale. Ordinal regression approaches can be applied to regression by discretizing the numeric range of the real-valued labels (Fu et al., 2018; Berg et al., 2021). In the existing works on ordinal regression by binary classification, $N - 1$ binary classifiers are used for target labels $\in \{1, 2, ..., N\}$, where classifier-$k$ predicts if the label is greater than $k$ or not for a given input. Li & Lin (2006) provided a reduction framework and generalization bound for the same. However, the proposed binary classification formulation is restricted. It requires several binary classifiers if the numeric range of output is extensive, whereas reducing the number of classifiers by using fewer quantization levels increases quantization error. Thus, a more generalized approach for using binary classification for regression is desirable to allow flexibility in the design of classifiers.

**Binary classification for multiclass classification:**  Dieterich & Bakiri (1995) proposed the use of error-correcting output codes (ECOC) to convert a multiclass classification to a set of binary classification problems. This improves accuracy as it introduces tolerance to binary classifiers' errors depending upon the hamming distance (i.e., number of bits changed between two binary strings) between two codes. Allwein et al. (2001) provided a unifying framework and multiclass loss bounds in terms of binary classification loss. More recent works have also used Hadamard code, a widely used error-correcting code (Song et al., 2021; Verma & Swami, 2019). Other works have focused on

Figure 1: The training (top) and inference (bottom) flow of binary-encoded labels (BEL) for regression networks. Red colored blocks represent design aspects we focus on.

the use and design of compact codes that exhibit a sublinear increase in the length of codes with the number of classes for extreme classification problems with a large number of classes (Cissé et al., 2012; Evron et al., 2018). However, the proposed encoding and decoding approaches do not consider the task objective and labels' ordinality for regression. Further, the binary classifiers possess distinct error probability distribution properties for regression problems as observed empirically (Section 3.1), which can be exploited to design codes suitable for regression.

Multiclass classification and ordinal regression by binary classification can be viewed as special cases falling under the BEL framework. As shown in Section 4, other BEL designs yield improvements in accuracy over these approaches. Task-specific regression techniques are well explored as summarized below (see also Appendix D). While effective, task-specific approaches lack generality by design.

**Head pose estimation:** SSR-Net (Yang et al., 2018) and FSA-Net (fsa, 2019) used a soft stagewise regression approach. HopeNet (Ruiz et al., 2018) used a combination of classification and regression loss. Hsu et al. (2019) used a combination of regression and ordinal regression loss.
**Facial landmark detection:** Wang et al. (2020) minimize L2 loss between predicted and target 2D heatmaps with the latter formed using small variance Gaussians centered on ground truth landmarks. AWing (Wang et al., 2019) modified loss for different pixels in the heatmap. LUVLi (Kumar et al., 2020) proposed a landmark's location, uncertainty, and visibility likelihood-based loss. Bulat & Tzimiropoulos (2016) used binary heatmaps with pixel-wise binary cross-entropy loss.
**Age estimation:** OR-CNN (Niu et al., 2016) and CORAL-CNN (Cao et al., 2020) used ordinal regression via binary classification. MV-Loss (Pan et al., 2018) proposed to penalize the model output based on the age distribution's variance, while Gao et al. (2018) proposed to use the KL-divergence between the softmax output and a generated label distribution for training.

## 3 BINARY-ENCODED LABELS FOR REGRESSION (BEL)

We consider regression problems where the goal is to minimize the error between real-valued target labels $y_i$ and predicted labels $\hat{y}_i$, over a set of training samples $i$. We transform this problem to a set of binary classification sub-problems by converting a real-valued label to a binary code.

Figure 1 shows the training and inference flow for BEL. The red-colored blocks highlight functions that vary under BEL. A real-valued label $y_i \in \mathbb{R}$ is quantized to level $Q_i \in \{1, 2, ..., N\}$ ①. The quantized label is converted to a binary vector $B_i \in \{0, 1\}^M$, that we call a *binary-encoded label*, using encoding function $\mathcal{E}$ ②. There are $\binom{2^M}{N}$ possible encoding functions—a large number. The binary-encoded labels $B_i$ are used to train $M$ classifiers ③. During inference the $M$ classifiers predict a binary code $\hat{B}_i \in \{0, 1\}^M$ for input $x_i$ ④. The predicted code ($\hat{B}_i$) or the predictions' magnitude ($\hat{Z}_i$), which indicates its confidence, is then decoded to a predicted label $\hat{y}_i \in \mathbb{R}$ using a decoding function $\mathcal{D}$ ⑤. We explore decoding functions that yield either quantized or continuous predicted outputs. The latter avoids quantization error by employing expected correlation (Section 3.3).

BEL contains five major design parameters resulting in a large design space: quantization, encoding, decoding, regressor network architecture, and training loss formulation. In this work we consider only uniform quantization while leaving nonuniform quantization (Fu et al., 2018) to future work. Section 3.2 and 3.3 explore the characteristics of suitable encoding, decoding, and loss functions. Section 3.4 explores the impact of regressor network architecture. We find varying any of these aspects can improve accuracy. While BEL provides a framework, and some design choices appear generally better than others, the most suitable BEL parameters to employ vary across task, dataset, and network architecture, as we show both theoretically (Section 3.1) and empirically (Section 4).

| $Q_i$ | $b^1b^2b^3b^4b^5b^6b^7$ | $b^1b^2b^3b^4$ |
|---|---|---|
| 1 | 0 0 0 0 0 0 0 | 0 0 0 0 |
| 2 | 1 0 0 0 0 0 0 | 0 0 0 1 |
| 3 | 1 1 0 0 0 0 0 | 0 0 1 1 |
| 4 | 1 1 1 0 0 0 0 | 0 1 1 1 |
| 5 | 1 1 1 1 0 0 0 | 1 1 1 1 |
| 6 | 1 1 1 1 1 0 0 | 1 1 1 0 |
| 7 | 1 1 1 1 1 1 0 | 1 1 0 0 |
| 8 | 1 1 1 1 1 1 1 | 1 0 0 0 |

| $b^1$ $b^2b^3$ |
|---|
| 0 00 |
| 0 01 |
| 0 11 |
| 0 10 |
| 1 00 |
| 1 01 |
| 1 11 |
| 1 10 |

| $Q_i$ | $b^1$ | $b^2b^3b^4b^5$ | $Q_i$ | $b^1$ | $b^2b^3b^4b^5$ |
|---|---|---|---|---|---|
| 1 | 0 | 0 0 0 0 | 9 | 1 | 1 0 0 0 |
| 2 | 0 | 0 0 0 1 | 10 | 1 | 1 1 0 0 |
| 3 | 0 | 0 0 1 1 | 11 | 1 | 1 1 1 0 |
| 4 | 0 | 0 1 1 1 | 12 | 1 | 1 1 1 1 |
| 5 | 0 | 1 1 1 1 | 13 | 1 | 0 1 1 1 |
| 6 | 0 | 1 1 1 0 | 14 | 1 | 0 0 1 1 |
| 7 | 0 | 1 1 0 0 | 15 | 1 | 0 0 0 1 |
| 8 | 0 | 1 0 0 0 | 16 | 1 | 0 0 0 0 |

| $Q_i$ | $b^1b^2$ | $b^3b^4$ | $Q_i$ | $b^1b^2$ | $b^3b^4$ |
|---|---|---|---|---|---|
| 1 | 0 0 | 0 0 | 9 | 1 1 | 0 0 |
| 2 | 0 0 | 0 1 | 10 | 1 1 | 0 1 |
| 3 | 0 0 | 1 1 | 11 | 1 1 | 1 1 |
| 4 | 0 0 | 1 0 | 12 | 1 1 | 1 0 |
| 5 | 0 1 | 1 0 | 13 | 1 0 | 1 0 |
| 6 | 0 1 | 1 1 | 14 | 1 0 | 1 1 |
| 7 | 0 1 | 0 1 | 15 | 1 0 | 0 1 |
| 8 | 0 1 | 0 0 | 16 | 1 0 | 0 0 |

| $Q_i$ | Hex($Q_i$) | $b^1b^2....b^8$ $b^9....b^{15}b^{16}$ |
|---|---|---|
| 0 | 00 | 00000000 00000000 |
| 1 | 01 | 00000000 00000001 |
| 2 | 02 | 00000000 00000011 |
| 15 | 0F | 00000000 10000000 |
| 32 | 20 | 00000011 00000000 |
| 33 | 21 | 00000011 00000001 |
| 34 | 22 | 00000011 00000011 |
| 35 | 23 | 00000011 00000111 |

(a)  (b) Unary  (c) Johnson  (d) B1JDJn  (e) B1JDJ  (f) B2JDJ  (g) HEXJ

Figure 2: Examples of BEL codes. Part (a) represents the quantized values of the labels for Unary and Johnson codes shown in Parts (b) and (c). Part (d) shows a B1JDJ code without reflected binary; Parts (e) and (f) show B1JDJ and B2JDJ codes for targets in the range 1 to 16. Part (g) shows quantized and encoded values for a HEXJ code (space added to differentiate between base and displacement, or digits). Red lines represent bit transitions. These BEL codes described in Section 3.2.

## 3.1 ANALYSIS OF ENCODING/DECODING FUNCTIONS

This section analyzes the potential impact of encoding/decoding functions on regression error assuming empirically observed error distributions for the underlying classifiers. We compare Unary and Johnson codes (Figure 2b and 2c) to determine when each is preferable. With this analytical study, we aim to obtain insight into ordinal label classifier impact on regression error when employing simple encoding and decoding functions $\{\mathcal{E}, \mathcal{D}\}$. Based upon this analysis we identify desirable properties for these functions. The design of the codes and intuition for trying them are discussed in Section 3.2. We divide our analysis into three parts: First, the expected error of predicted labels is derived in terms of classifiers' errors for two $\{\mathcal{E}, \mathcal{D}\}$ functions. Next, we propose an approximate classifier's error probability distribution over the numeric range of target labels for regression based on empirical study. Last, we compare the expected error of sample $\{\mathcal{E}, \mathcal{D}\}$ functions based on our analysis. We use labels $y_i \in [1, N-1]$, with quantization levels $Q_i \in \{1, 2, ..., N-1\}$. Quantization error is not included as it is not affected by $\{\mathcal{E}, \mathcal{D}\}$ functions.

**Expected absolute error bounds in terms of classification error:**   First, we analyze the unary code (BEL-U). The encoding function $\mathcal{E}^{\text{BEL-U}}$ converts $Q_i$ to $B_i = b_i^1, b_i^2, ..., b_i^{N-2}$, where $b_i^k = 1$ for $k < Q_i$, else 0. In this case, a good choice of decoding function turns out to be simply counting the number of 1 outputs across all $N-2$ classifiers since a error in a single classifier changes the prediction by only one quantization level. Adding one since $Q_i = 1$ is encoded by all zeros gives:

$$\mathcal{D}^{\text{BEL-U}}(\hat{b}_i^1, \hat{b}_i^2, ..., \hat{b}_i^{N-2}) = \sum_{k=1}^{N-2} \hat{b}_i^k + 1 \tag{1}$$

Let $e_k(n)$ be the error probability of classifier $k$ for target quantized label $Q_i = n$. For a uniform distribution of $y_i$ in the range $[1, N-1]$ the expected error for BEL-U can be shown (see Appendix B) to be bounded as follows:

$$\mathbb{E}(|\hat{y}^{\text{BEL-U}} - y|) \leqslant \frac{1}{N-1} \sum_{n=1}^{N-1} \left( \sum_{k=1}^{N-2} e_k(n) \right) \tag{2}$$

A similar analysis of expected error can be applied to binary encoded labels constructed to yield Johnson codes (BEL-J), in which $Q_i$ is encoded using $B_i = b_i^1, b_i^2, ..., b_i^{N/2}$, where, $b_i^k = 1$ for $\frac{N}{2} - Q_i < k - 1 \leqslant N - Q_i$, else 0 (see Equation 27 in Appendix B).

**Error probability of classifiers:**   To use Equation 2 we need to determine $e_k(n)$. A classifier's target output is 0 or 1. For BEL, the target labels of a given classifier will have one or more *bit transitions* from $0 \to 1$ or $1 \to 0$ as the target value of the regression network's output varies. For example, for the unary code (Figure 2b), the target output of classifier $b^2$ has a bit transition from 0 to 1 going from $Q_i = 2$ to $Q_i = 3$. The classifier should learn a decision boundary in $(2, 3)$. Each BEL classifier is tasked with learning decision boundaries for all bit transitions. As the difficulty of this task varies with the number of bit transitions it varies with different encoding functions. Moreover, the misclassification probability of a classifier tends to increase as the target label is closer to the

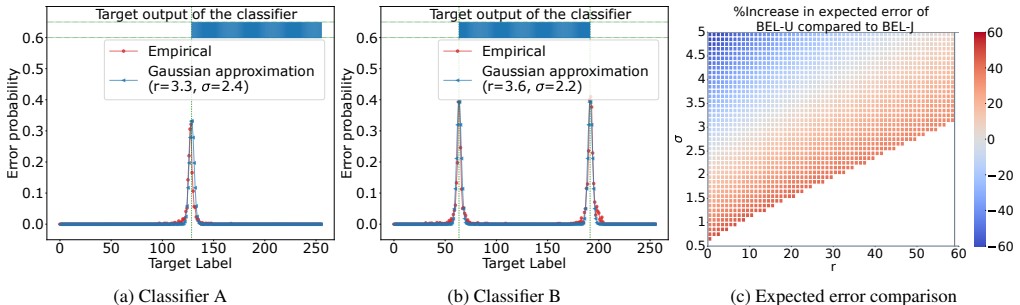

(a) Classifier A  (b) Classifier B  (c) Expected error comparison

Figure 3: Part (a) and (b): classification error probability vs. target output for two classifiers. Target output 1 where blue and 0 elsewhere. Part (c): expected error increase of BEL-U versus BEL-J based on Equation 2 to Equation 4 (blank means that combination of $r$ and $\sigma$ results in an error probability greater than one).

classifier's decision boundaries (Cardoso & Pinto da Costa, 2007). Thus, we approximate $e_k(y)$ for a classifier $k$ with $t$ bit transitions as a linear combination of $t$ Gaussian distributions. Here, each Gaussian term is centered around a bit transition. Let $f_{\mathcal{N}(\mu,\sigma^2)}(y)$ denote the probability density of a normal distribution with mean $\mu$ and variance $\sigma^2$. Each classifier for BEL-U encoding has one bit transition, whereas, each classifier for BEL-J encoding has two bit transitions (except the first and last classifiers). $e_k(y)$ of a classifier $k$ for BEL-U and BEL-J encoding is approximated as:

$$e_k^{\text{BEL-U}}(y) = r f_{\mathcal{N}(\mu_k,\sigma^2)}(y), \text{where}, \mu_k = k + 0.5 \tag{3}$$

$$e_k^{\text{BEL-J}}(y) = r f_{\mathcal{N}(\mu_{1k},\sigma^2)}(y) + r f_{\mathcal{N}(\mu_{2k},\sigma^2)}(y), \text{where}, \mu_{1k} = \frac{N}{2} - k + 1.5, \mu_{2k} = N - k + 1.5 \tag{4}$$

Here, $r$ is a scaling factor. Figure 3a and 3b compares Equation 3 and 4 against empirically observed error distributions for two classifiers using an HRNetV2-W18 (Wang et al., 2020) feature extractor (backbone) trained with the COFW facial landmark detection dataset (Burgos-Artizzu et al., 2013).

**Comparison of expected absolute error for BEL-U and BEL-J:**     Based on the above analysis, we compare the expected absolute errors of BEL-U and BEL-J. Figure 3c represents the percentage increase in absolute error for BEL-U compared to BEL-J for valid values of standard deviation $\sigma$ ($y-$axis) and scaling factor $r$ ($x-$axis) as used in Equation 3 and 4. Here, BEL-J has a lower error in the red-colored region (%increase$> 0$), whereas BEL-U has a lower error in the blue-colored region (%increase $< 0$). The figure shows that whether BEL-J or BEL-U has lower error depends upon the values of $\sigma$ and $r$. This suggests that the best $\{\mathcal{E}, \mathcal{D}\}$ function will depend upon the classifier error probability distribution. The classifier error distribution in turn may depend upon the task, dataset, label distribution, network architecture, and optimization approach. Derivation of expected error for BEL-U and BEL-J and classifiers' empirical error probability distributions for different architectures, datasets, and encodings are provided in Appendix B to C.

## 3.2 DESIGN OF ENCODING FUNCTIONS

Based on the above analysis and further empirical observation we identify three principles for selecting BEL codes for regression so as to minimize error. First, *individual classifiers should require fewer bit transitions* as this makes them easier to train. Second, a desirable property for a BEL encoding function is that *the hamming distance between two codes (number of bits that differ) should be proportional to the difference between the target values they encode.* However, hamming distance weighs all bit changes equally. Thus, hamming distance based code design provides equal error protection capability to all bits (Wu, 2018; Xie et al., 2002) and does not account for which classifiers are more likely to mispredict for a given input. This matters because the misclassification probability of BEL classifiers is not uniform, but rather increases the closer the target value of an input is to a bit transition  (e.g., Figure 3a and 3b). These observations yield a third important consideration:  *For a given target value classifiers closer to a bit transition are more likely to incur an error.*

The principles above highlight a tradeoff between classification error probability and error-correction properties when selecting BEL codes. To evaluate the trade-offs, we empirically evaluate encodings that, to greater or lesser extent, satisfy one or more of the principles while focusing on reducing the number of classifiers (bits) so as to avoid increasing model parameters. Development of algorithms

that might systematically optimize encoding functions is left to future work. Specifically, we explore the following codes:

**Unary code (U):** Unary codes (Figure 2b) have only one bit transition per classifier and thus require $M = N - 1$ bits to encode $N$ values. Unary codes satisfy the first two principles and prior work on ordinal regression by binary classification (Li & Lin, 2006; Niu et al., 2016) uses similar codes.

**Johnson code (J):** The Johnson code sequence (Figure 2c) is based on Libaw-Craig code (Libaw & Craig, 1953). We select this code as it has well-separated bit transitions and requires $M = \frac{N}{2}$ bits compared to $N$ required for unary code. This code exemplifies the impact of considering non-uniform classifier error probabilities (third principle). For example the hamming distance between 1 and 8 is just one. However, the bit transition for the differing bit, for classifier $b^1$, is far from 1 or 8. Assuming equal error probability distributions centered on each bit transition for each classifier (as in Equation 4), $b^1$ is less likely to mispredict than $b^2$, $b^3$ or $b^4$ for inputs with target values near 1 or 8.

**Base+displacement based code (B1JDJ/B2JDJ):** We further reduce the number of bits using a base+displacement-based representation. In this representation, a value is represented in base-k using a base-term `b` and displacement `d` via `b * k + d`. `b` and `d` are represented using Johnson codes. Further, to improve the distance between two remote codes, we adapt reflected binary codes for term `d` (Gray, 1953). We evaluate base-2 (B1JDJ - Figure 2e) and base-4 codes (B2JDJ - Figure 2f).

**Binary coded hex - Johnson code (HEXJ):** In HEXJ (Figure 2g), each digit (0-F) of the hexadecimal representation of a number is converted to an 8-bit binary code using Johnson code. For example, for the decimal number 47 (i.e., 2F in hex), HEXJ(47) = Concatanate(Johnson(2), Johnson(F)). A 16-bit HEXJ code can represent numbers in the range of 00 to FF (a total of 256). The number of bits increases sublinearly with the number of quantization levels for HEXJ, making it suitable for regression problems with many quantization levels.

**Hadamard code (HAD):** Hadamard codes (Bose & Shrikhande, 1959) are widely used as error-correcting codes and have been used for multiclass classification (Dietterich & Bakiri, 1995; Verma & Swami, 2019). They require $M = N$ bits to encode $N$ values. However, Hadamard codes violate all three BEL code selection principles: First, each classifier has many bit transitions. Second, as each code is equidistant (hamming distance of $\frac{M}{2}$), the difference between target values is ignored. Finally, they protect all bits equally so do not take advantage of non-uniform error probabilities. We verify empirically Hadamard codes are unsuitable for regression (Section A).

### 3.3 Design of Decoding Functions

We explore three decoding functions: custom decoding, correlation-based decoding, and expected correlation-based decoding. Custom decoding functions are specific to the encoding function, and are only evaluated for unary and Johnson codes. In contrast, correlation-based decoding, first explored in prior work studying ECOC for multiclass classification (Allwein et al., 2001), can be applied to all codes. For quantized labels in $\{1, 2, ..., N\}$, we define a code matrix $C$ of size $N \times M$, where $M$ is the number of bits/classifiers used for the binary-encoded label. Each row $C_{k,:}$ in this matrix represents the binary code for label $Q_i = k$. For example, Figure 2b can be considered a code matrix, where the first row represents code for label $Q_i = 1$. Let $\hat{Z}_i \in \mathbb{R}^M$ denote the output logit values of the classifiers. For decoding, the row with the highest correlation with the output $\hat{Z}_i$ is selected as the decoded label. Here, real-valued output $\hat{Z}_i$ is used instead of output binary code $\hat{B}_i$ to find the correlation as it uses the confidence of a classifier to make a more accurate prediction. For target quantized labels $Q_i \in \{1, 2, ..., N\}$, the decoding function is defined as:

$$\mathcal{D}^{\text{GEN}}(\hat{Z}_i) = \underset{k \in \{1,2,...,N\}}{\operatorname{argmax}} \left( \hat{Z}_i \cdot C_{k,:} \right) \tag{5}$$

However, $\mathcal{D}^{\text{GEN}}$ outputs a quantized prediction, introducing quantization error. To remedy this concern and demonstrate the potential of more sophisticated decoding rules, we propose and evaluate an expected correlation-based decoding function, which allows prediction of real-valued label $\hat{y}_i$. For target labels $y_i \in [1, N]$, the decoding function is defined as:

$$\mathcal{D}^{\text{GEN-EX}}(\hat{Z}_i) = \sum_{k=1}^{N} k\sigma_k, \text{ where } \sigma_k = \frac{e^{\hat{Z}_i \cdot C_{k,:}}}{\sum_{j=1}^{N} e^{\hat{Z}_i \cdot C_{j,:}}} \tag{6}$$

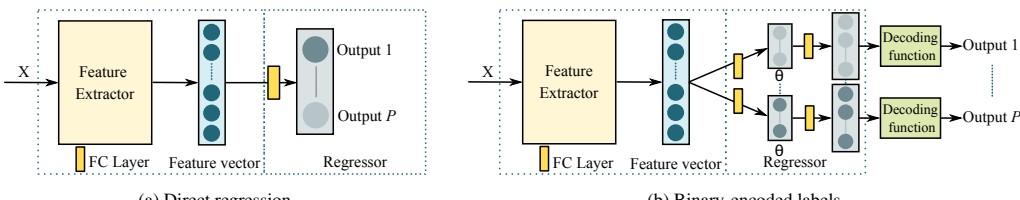

Figure 4: Network architecture for direct and BEL regression; only the regressor architecture is modified, but the entire network is trained end to end. $P$ is the number of dimensions of the regression network output.

Table 1: Benchmarks used for evaluation

| Task | Feature Extractor | Specialized Approach | Dataset | Benchmark | Label range/ Quantization levels | $\theta$ |
|---|---|---|---|---|---|---|
| Landmark-free 2D head pose estimation | ResNet50 | Regression+classification (Ruiz et al., 2018) | BIWI | HPE1 | -100-100/200 | 10 |
| | | | 300LP/AFLW2000 | HPE2 | -100-100/200 | 10 |
| | RAFANet | Direct regression (Behera et al., 2021) | BIWI | HPE3 | -180-180/360 | 50 |
| | | | 300LP/AFLW2000 | HPE4 | -180-180/360 | 50 |
| Facial Landmark Detection | HRNetV2-W18 | Heatmap regression (Wang et al., 2020; Xu et al., 2020) | COFW | FLD1 | 0-256/256 | 10 |
| | | | 300W | FLD2 | 0-256/256 | 10 |
| | | | WFLW | FLD3 | 0-256/256 | 10 |
| | | | AFLW | FLD4 | 0-256/256 | 30 |
| Age estimation | ResNet50 /ResNet34 | Ordinal regression (Cao et al., 2020) | MORPH-II | AE1 | 0-64/64 | 10 |
| | | | AFAD | AE2 | 0-32/32 | 10 |
| End-to-end autonomous driving | PilotNet | Direct regression (Bojarski et al., 2017) | PilotNet | PN | 0-670/670 | 10 |

**Training loss functions:** A deep neural network with multiple output binary classifiers can be trained using the binary cross-entropy (BCE) loss $\mathcal{L}_{\mathrm{BCE}}\big(\hat{Z}_i, \mathcal{E}(Q_i)\big)$. However, this loss minimizes the mismatch between predicted and target code but does not directly minimize the error between the target and predicted values. Decoding functions $\mathcal{D}^{\mathrm{GEN}}$ and $\mathcal{D}^{\mathrm{GEN\text{-}EX}}$ can be used to calculate the loss and minimize the mismatch between decoded predictions and target values directly. Decoding function $\mathcal{D}^{\mathrm{GEN}}$ finds the correlation between each row of the code matrix ($\boldsymbol{C}_{k,:}$) and the output $\hat{Z}_i$. $\boldsymbol{C}\hat{Z}_i$ gives the correlation vector, and the index with the highest correlation is used as the predicted label. In this case, cross-entropy loss $\mathcal{L}_{\mathrm{CE}}\big(\boldsymbol{C}\hat{Z}_i, Q_i\big)$ can be used to train the network. Similarly, for decoding function $\mathcal{D}^{\mathrm{GEN\text{-}EX}}$, which predicts a continuous value, L1 or L2 loss $\mathcal{L}_{\mathrm{L1/L2}}\big(\mathcal{D}^{\mathrm{GEN\text{-}EX}}(\hat{Z}_i), y_i\big)$ can also be used for training. We evaluate multiple combinations of decoding and loss functions in Section 4.

### 3.4 REGRESSION NETWORK ARCHITECTURE FOR BEL

A regression network typically consists of a feature extractor and regressor. The regressor consists of a fully connected layer between the feature extractor's output (i.e., feature vector) and output logits for direct regression as shown in Figure 4a. In BEL, the number of output logits is increased to the number of classifiers (bits) used. When $y \in \mathbb{R}^P$, with $P > 1$, the required number of output logits—$P \times M$ assuming $M$-bit BEL encoding per output dimension—can significantly increase the size of the regression layer. However, empirically, we find small feature vectors suffice as the output logits are highly correlated for the explored encoding functions. Adding a fully connected bottleneck layer to reduce feature vector size to $\theta$ reduces the number of parameters and provides a trade-off between the model size and accuracy. Figure 4b shows the modified network architecture for BEL.

## 4 EVALUATION

Table 1 summarizes the tasks, datasets, and network architectures used for the evaluation of BEL. These tasks are commonly used for evaluation of regression approaches by prior works due to the complexity of problem and network architectures (Díaz & Marathe, 2019). Landmark-free 2D head pose estimation (HPE) aims to find a human head's pose in terms of three angles: yaw, pitch, and roll from a 2D image without landmarks. Facial landmark detection (FLD) is a problem of detecting the $(x, y)$ coordinates of keypoints in a given face image. Age estimation aims to predict the age of a person from an image. In end-to-end autonomous driving, the steering wheel's next angle is predicted

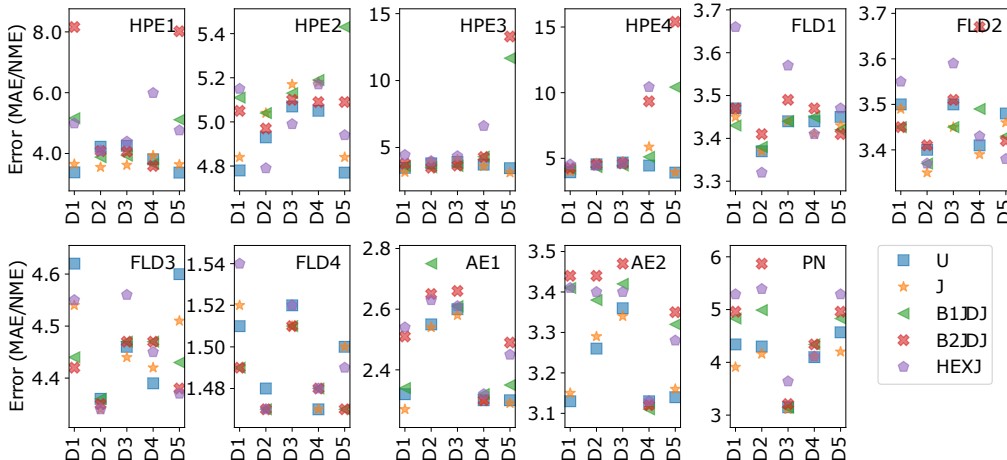

Figure 5: Error (MAE or NME) for different encoding, decoding, and loss functions for BEL. D1-D5 represents different combinations of decoding and loss functions: D1 (BCE loss with BEL-U/BEL-J/GEN decoding for U/J/others), D2 (CE/GEN-EX), D3 (CE/GEN), D4 (L1 or L2/GEN-EX), and D5 (BCE/GEN-EX).

from an image of the road. Normalized Mean Error (NME) and Mean Absolute Error (MAE) with respect to raw real-valued labels are used as the evaluation metric for FLD and the rest, respectively.

We also evaluate direct regression and multiclass classification as baseline regression approaches. For direct regression, L1 or L2 loss functions are used. Label values are scaled to reduce the range of labels. The loss function and the scaling factors are set using hyperparameter tuning. In the multiclass classification-based regression, the target values are quantized and converted to a class. The network is trained using cross-entropy loss in this case. In our evaluation, the entire network (i.e., feature extractor and regressor) are trained end-to-end for direct regression, multiclass classification, and BEL. The feature extractor, data augmentation, evaluation protocols, and the number of training iterations are kept uniform across different methods for each benchmark. We report average of five training runs and error margin of $95\%$ confidence interval. Details on datasets, training parameters, related work for specific tasks, and other evaluation metrics are provided in Appendix C.

BEL introduces several design parameters for regression by binary classification. We evaluate different encoding ($\mathcal{E}$), decoding ($\mathcal{D}$), and training loss ($\mathcal{L}$) functions for BEL across all the benchmarks and study the extent and nature of the impact of these design parameters on accuracy.

**Encoding function ($\mathcal{E}$):** Figure 5 plots error (MAE or NME) using different encodings. We do not show results for Hadamard codes here as it results in significantly higher error than other encodings (Appendix A). On average, Hadamard codes result in $\sim 60\%$ higher error than J encoding, which shows that these codes are unsuitable for regression. The results show the encoding function significantly affects the accuracy and the best-performing encoding function varies across tasks, datasets, and network architectures (e.g., HPE1 and HPE3 are trained on the same dataset and different architecture). In Section 3.1 we observed that which encoding/decoding functions result in lower error depends upon the classifiers' error distribution. For decoding functions used for the comparison in Section 3.1, J does better than U for HPE3, FLD1, and AE1; we attribute this to misclassification errors occurring more frequently near bit transitions based on the analytical study.

The encoding function impacts the number of classifiers and the complexity of the function to be learned by a classifier. We observe a trade-off between these two parameters. For some benchmarks, the availability of sufficient training data and network capacity facilitates the learning of complex classifiers such as B2JDJ. In such a case, a reduced number of classifiers compared to U, J, or B1JDJ codes results in a lower error. We provide empirical results for the same in Appendix A.

**Decoding ($\mathcal{D}$) and training loss ($\mathcal{L}$) functions:** We explore three decoding and three training loss functions (Section 3.3). However, not all the combinations of decoding and loss functions ($\mathcal{D}/\mathcal{L}$) perform well. For example, CE, L1, or L2 losses do not use decodings $\mathcal{D}^{\text{BEL-J}}$ or $\mathcal{D}^{\text{BEL-U}}$. Therefore, optimizing the network for these losses does not directly minimize the absolute error between targets

Table 2: Comparison of BEL with different regression approaches. "Specialized approach" described in Table 1

| Approach | Error (MAE or NME) / Model size | | | |
| --- | --- | --- | --- | --- |
| | HPE1 | HPE2 | HPE3 | HPE4 |
| Specialized approach | - | - | 3.40 / 69.8M | 4.14 / 69.8M |
| Direct regression | 4.76 ±0.35 / 23.5M | 5.65 ±0.13 / 23.5M | 3.40 ±0.26 / 69.8M | 4.14 ±0.12 / 69.8M |
| Multiclass classification | 4.49 ±0.24 / 24.2M | 5.31 ±0.05 / 24.8M | 4.54 ±0.04 / 72.0M | 5.14 ±0.08 / 72.0M |
| BEL | **3.56**±0.01 / 23.6M | **4.77** ±0.05 / 23.6M | **3.30**±0.04 / 69.8M | **3.90** ±0.03 / 69.8M |
| BEL $\mathcal{E}/\mathcal{D}/\mathcal{L}$ functions | U/GEN-EX/L2 | U/GEN-EX/BCE | B1JDJ/GEN-EX/BCE | U/GEN-EX/BCE |
| Approach | FLD1 | FLD2 | FLD3 | FLD4 |
| Specialized approach | 3.45 / 9.6M | **3.32** / 9.6M | **4.32** / 9.6M | 1.57 / 9.6M |
| Direct regression | 3.60 ±0.02 / 10.2M | 3.54 ±0.03 / 10.2M | 4.64 ±0.03 / 10.2M | 1.51 ±0.01 / 10.2M |
| Multiclass classification | 3.58 ±0.03 / 25.4M | 3.51 ±0.02 / 45.2M | 4.50 ±0.01 / 61.3M | 1.56 ±0.01 / 20.1M |
| BEL | **3.34** ±0.02 / 10.6M | 3.40 ±0.02 / 11.2M | 4.36 ±0.02 / 11.7M | **1.47** ±0.00 / 10.8M |
| BEL $\mathcal{E}/\mathcal{D}/\mathcal{L}$ functions | HEXJ/GEN-EX/CE | U/GEN-EX/CE | B1JDJ/GEN-EX/CE | B1JDJ/GEN-EX/CE |
| Approach | AE1 | AE2 | PN | |
| Specialized approach | 2.49 / 21.3M | 3.47 / 21.3M | 4.24 / 1.8M | |
| Direct regression | 2.44 ±0.01 / 23.1M | 3.21 ±0.02 / 23.1M | 4.24 ±0.45 / 1.8M | |
| Multiclass classification | 2.75 ±0.03 / 23.1M | 3.38 ±0.05 / 23.1M | 5.54 ±0.00 / 1.9M | |
| BEL | **2.27** ±0.01 / 23.1M | **3.11** ±0.00 / 23.1M | **3.11** ±0.01 / 1.8M | |
| BEL $\mathcal{E}/\mathcal{D}/\mathcal{L}$ functions | J/BEL-J/BCE | B1JDJ/GEN-EX/L1 | J/GEN/CE | |

and decoded predictions. We present results for five out of nine $\mathcal{D}/\mathcal{L}$ combinations. Figure 5 compares error (MAE or NME) achieved by different $\mathcal{D}/\mathcal{L}$ combinations and highlights the range of error variations. $\mathcal{D}^{\text{GEN-EX}}$ results in the lowest error for the majority of the benchmarks as it reduces quantization error and also utilizes the output logit confidence values. $\mathcal{D}^{\text{GEN-EX}}$ consistently perform better than $\mathcal{D}^{\text{GEN}}$ function that has been used for multiclass classification by prior works (Allwein et al., 2001). The use of CE or L1/L2 loss results in a lower error with $\mathcal{D}^{\text{GEN-EX}}$ for most benchmarks as the training loss function directly minimizes the error between targets and decoded predictions.

**Comparison of BEL with regression approaches:** Table 2 compares BEL with other approaches for different benchmarks (Table 1). We explore and evaluate multiple combinations of encoding ($\mathcal{E}$), decoding ($\mathcal{D}$), and loss ($\mathcal{L}$) functions for BEL in this work. In these experiments 20% of the training set is used as validation set and the validation error is used to choose the best BEL approach. An ablation study for using more fully connected layers for direct regression and multiclass classification is in Appendix A. BEL results in lower error than direct regression and multiclass classification and even outperforms task-specific regression approaches for several benchmarks.

*The results show no single combination of encoding/decoding/loss functions evaluated was best for all benchmarks but also demonstrate BEL improves accuracy across a range of regression problems.*

## 5 CONCLUSION

This work proposes binary-encoded labels (BEL) to pose regression as binary classification. We propose a taxonomy identifying the key design aspects for regression by binary classification and demonstrate the impact of classification error and encoding/decoding functions on the expected label error. Different encoding, decoding, and loss functions are explored to evaluate our approach using four complex regression tasks. BEL results in an average 9.9%, 15.5%, and 7.2% lower error than direct regression, multiclass classification, and task-specific regression approaches, respectively. BEL improves accuracy over state-of-the-art approaches for head pose estimation (BIWI, AFLW2000), facial landmark detection (COFW), age estimation (AFAD), and end-to-end autonomous driving (PilotNet). Our analysis and empirical evaluation in this work demonstrate the potential of the vast design space of BEL for regression problems and the importance of finding suitable design parameters for a given task. The best performing encoding/decoding function pair may be task, dataset, and network specific. A possibility this suggests, which we leave to future work, is that it may be beneficial to develop automated approaches for optimizing these functions.

## 6 ACKNOWLEDGEMENTS

This research has been funded in part by the National Sciences and Engineering Research Council of Canada (NSERC) Strategic Project Grant. Tor M. Aamodt serves as a consultant for Huawei Technologies Canada Co. Ltd. and Intel Corp. Deval Shah is partly funded by the Four Year Doctoral Fellowship (4YF) provided by the University of British Columbia.

**Reproducibility:** We have provided a detailed discussion about training hyperparameters, experimental setup, and modifications made in publicly available network architectures in Appendix D.1-D.4 for all benchmarks. Code is available at `https://github.com/ubc-aamodt-group/BEL_regression`. We have provided the training and inference code with trained models.

**Code of Ethics** Some of the major applications of regression problems are artificial intelligence and autonomous machines, and regression improvement can accelerate the development of autonomous systems. However, depending upon the use, autonomous systems can have some negative societal impacts, such as job loss in some sectors and ethical concerns.

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

## A    ABLATION STUDY

**Impact of combination of encoding, decoding, and loss functions:**    We propose multiple combinations of encoding, decoding, and loss functions that can be used with BEL. In Tables 3- 13, we show the effect of each combination of encoding, decoding, and loss function on the error of the model. Although general trends exist and some combinations perform consistently well across datasets, the optimal combination varies based on the dataset.

Table 3: Comparison of BEL design parameters on MAE for head pose estimation with BIWI dataset and ResNet50 feature extractor (HPE1).

| Decoding function | Loss function | Encoding function | | | | | |
|---|---|---|---|---|---|---|---|
| | | U | J | B1JDJ | B2JDJ | HEXJ | HAD |
| BEL-J/BEL-U | BCE | 3.38 | 3.65 | - | - | - | - |
| GEN-EX | BCE | **3.37** | 3.64 | 5.11 | 8.02 | 4.76 | 7.53 |
| GEN | BCE | 3.38 | 3.65 | 5.16 | 8.16 | 4.99 | 7.73 |
| GEN-EX | CE | 4.22 | 3.55 | 3.88 | 4.08 | 4.09 | 5.50 |
| GEN | CE | 4.25 | 3.62 | 3.93 | 4.06 | 4.39 | 5.48 |
| GEN-EX | L2 | 3.56 | 3.93 | 3.66 | 3.59 | 5.99 | 4.21 |

Table 4: Comparison of BEL design parameters on MAE for head pose estimation with 300LP/AFLW2000 datasets and ResNet50 feature extractor (HPE2).

| Decoding function | Loss function | Encoding function | | | | | |
|---|---|---|---|---|---|---|---|
| | | U | J | B1JDJ | B2JDJ | HEXJ | HAD |
| BEL-J/BEL-U | BCE | 4.78 | 4.84 | - | - | - | - |
| GEN-EX | BCE | **4.77** | 4.84 | 5.43 | 5.09 | 4.94 | 7.84 |
| GEN | BCE | 4.78 | 4.87 | 5.11 | 5.05 | 5.15 | 8.54 |
| GEN-EX | CE | 4.93 | 5.04 | 5.04 | 4.97 | 4.79 | 5.64 |
| GEN | CE | 5.07 | 5.17 | 5.13 | 5.10 | 4.99 | 5.62 |
| GEN-EX | L2 | 5.05 | 5.18 | 5.19 | 5.09 | 5.17 | 5.07 |

Table 5: Comparison of BEL design parameters on MAE for head pose estimation with BIWI dataset and RAFA-Net feature extractor (HPE3).

| Decoding function | Loss function | Encoding function | | | | | |
|---|---|---|---|---|---|---|---|
| | | U | J | B1JDJ | B2JDJ | HEXJ | HAD |
| BEL-J/BEL-U | BCE | 3.47 | 3.16 | - | - | - | - |
| GEN-EX | BCE | 3.46 | **3.12** | 3.30 | 3.35 | 3.80 | 5.75 |
| GEN | BCE | 3.49 | 3.14 | 3.62 | 3.78 | 4.44 | 5.83 |
| GEN-EX | CE | 3.82 | 3.91 | 3.52 | 3.49 | 3.98 | 3.98 |
| GEN | CE | 3.92 | 4.09 | 3.62 | 3.65 | 4.35 | 4.28 |
| GEN-EX | L2 | 3.72 | 3.60 | 4.31 | 4.29 | 6.61 | 18.69 |

Table 6: Comparison of BEL design parameters on MAE for head pose estimation with 300LP/AFLW2000 datasets and RAFA-Net feature extractor (HPE4).

| Decoding function | Loss function | Encoding function | | | | | |
|---|---|---|---|---|---|---|---|
| | | U | J | B1JDJ | B2JDJ | HEXJ | HAD |
| BEL-J/BEL-U | BCE | 3.94 | 4.00 | - | - | - | - |
| GEN-EX | BCE | **3.90** | 3.93 | 4.19 | 4.12 | 4.39 | 9.17 |
| GEN | BCE | 3.93 | 3.94 | 4.21 | 4.25 | 4.53 | 9.21 |
| GEN-EX | CE | 4.55 | 4.62 | 4.34 | 4.53 | 4.45 | 5.12 |
| GEN | CE | 4.68 | 4.75 | 4.46 | 4.61 | 4.63 | 5.29 |
| GEN-EX | L2 | 4.45 | 5.87 | 5.11 | 9.34 | 10.43 | 17.89 |

Table 7: Comparison of BEL design parameters on NME for facial landmark detection with COFW dataset and HRNetV2-W18 feature extractor (FLD1).

| Decoding function | Loss function | Encoding function | | | | | |
|---|---|---|---|---|---|---|---|
| | | U | J | B1JDJ | B2JDJ | HEXJ | HAD |
| BEL-J/BEL-U | BCE | 3.47 | 3.45 | - | - | - | - |
| GEN-EX | BCE | 3.45 | 3.43 | 3.42 | 3.41 | 3.47 | 4.28 |
| GEN | BCE | 3.46 | 3.45 | 3.43 | 3.47 | 3.66 | 4.43 |
| GEN-EX | CE | 3.37 | 3.37 | 3.38 | 3.41 | **3.34** | 3.69 |
| GEN | CE | 3.44 | 3.44 | 3.44 | 3.49 | 3.57 | 3.69 |
| GEN-EX | L1 | 3.44 | 3.41 | 3.45 | 3.47 | 3.41 | 4.52 |

Table 8: Comparison of BEL design parameters on NME for facial landmark detection with 300W dataset and HRNetV2-W18 feature extractor (FLD2).

| Decoding function | Loss function | Encoding function | | | | | |
|---|---|---|---|---|---|---|---|
| | | U | J | B1JDJ | B2JDJ | HEXJ | HAD |
| BEL-J/BEL-U | BCE | 3.5 | 3.49 | - | - | - | - |
| GEN-EX | BCE | 3.48 | 3.46 | 3.43 | 3.42 | 3.38 | 4.71 |
| GEN | BCE | 3.50 | 3.49 | 3.45 | 3.45 | 3.55 | 4.78 |
| GEN-EX | CE | 3.40 | **3.36** | 3.37 | 3.41 | 3.37 | 3.62 |
| GEN | CE | 3.50 | 3.45 | 3.45 | 3.51 | 3.59 | 3.65 |
| GEN-EX | L1 | 3.41 | 3.39 | 3.49 | 3.67 | 3.43 | 4.04 |

Table 9: Comparison of BEL design parameters on NME for facial landmark detection with WFLW dataset and HRNetV2-W18 feature extractor (FLD3).

| Decoding function | Loss function | Encoding function | | | | | |
|---|---|---|---|---|---|---|---|
| | | U | J | B1JDJ | B2JDJ | HEXJ | HAD |
| BEL-J/BEL-U | BCE | 4.62 | 4.54 | - | - | - | - |
| GEN-EX | BCE | 4.6 | 4.51 | 4.43 | 4.38 | 4.37 | 7.18 |
| GEN | BCE | 4.62 | 4.53 | 4.44 | 4.42 | 4.55 | 7.14 |
| GEN-EX | CE | 4.36 | 4.34 | 4.36 | **4.33** | 4.34 | 5.15 |
| GEN | CE | 4.46 | 4.44 | 4.47 | 4.47 | 4.56 | 4.83 |
| GEN-EX | L1 | 4.39 | 4.42 | 4.47 | 4.47 | 4.45 | 4.74 |

Table 10: Comparison of BEL design parameters on NME for facial landmark detection with AFLW dataset and HRNetV2-W18 feature extractor (FLD4).

| Decoding function | Loss function | Encoding function | | | | | |
|---|---|---|---|---|---|---|---|
| | | U | J | B1JDJ | B2JDJ | HEXJ | HAD |
| BEL-J/BEL-U | BCE | 1.51 | 1.52 | - | - | - | - |
| GEN-EX | BCE | 1.50 | 1.50 | **1.47** | **1.47** | 1.49 | 1.52 |
| GEN | BCE | 1.51 | 1.52 | 1.50 | 1.49 | 1.54 | 1.55 |
| GEN-EX | CE | 1.48 | **1.47** | **1.47** | **1.47** | **1.47** | **1.47** |
| GEN | CE | 1.52 | 1.51 | 1.51 | 1.51 | 1.52 | 1.52 |
| GEN-EX | L1 | **1.47** | **1.47** | 1.48 | 1.48 | 1.48 | 1.59 |

Table 11: Comparison of BEL design parameters on MAE for age estimation with MORPH-II dataset and ResNet50 feature extractor (AE1).

| Decoding function | Loss function | Encoding function | | | | | |
|---|---|---|---|---|---|---|---|
| | | U | J | B1JDJ | B2JDJ | HEXJ | HAD |
| BEL-J/BEL-U | BCE | 2.32 | **2.27** | - | - | - | - |
| GEN-EX | BCE | 2.30 | 2.29 | 2.35 | 2.49 | 2.45 | 2.99 |
| GEN | BCE | 2.28 | 2.28 | 2.34 | 2.51 | 2.54 | 3.07 |
| GEN-EX | CE | 2.55 | 2.54 | 2.75 | 2.65 | 2.63 | 12.33 |
| GEN | CE | 2.60 | 2.58 | 2.61 | 2.66 | 2.61 | 3.10 |
| GEN-EX | L1 | 2.30 | 2.30 | 2.32 | 2.30 | 2.32 | 2.29 |

Table 12: Comparison of BEL design parameters on MAE for age estimation with AFAD dataset and ResNet50 feature extractor (AE2).

| Decoding function | Loss function | Encoding function | | | | | |
|---|---|---|---|---|---|---|---|
| | | U | J | B1JDJ | B2JDJ | HEXJ | HAD |
| BEL-J/BEL-U | BCE | 3.13 | 3.15 | - | - | - | - |
| GEN-EX | BCE | 3.14 | 3.16 | 3.32 | 3.35 | 3.28 | 3.34 |
| GEN | BCE | 3.13 | 3.19 | 3.41 | 3.44 | 3.41 | 3.52 |
| GEN-EX | CE | 3.26 | 3.29 | 3.38 | 3.44 | 3.40 | 3.30 |
| GEN | CE | 3.36 | 3.34 | 3.42 | 3.47 | 3.40 | 3.45 |
| GEN-EX | L1 | 3.13 | 3.12 | **3.11** | 3.12 | 3.13 | 3.13 |

Table 13: Comparison of BEL design parameters on MAE for end-to-end autonomous driving with PilotNet dataset and feature extractor (PN).

| Decoding function | Loss function | Encoding function | | | | | |
|---|---|---|---|---|---|---|---|
| | | U | J | B1JDJ | B2JDJ | HEXJ | HAD |
| BEL-J/BEL-U | BCE | 4.34 | 3.91 | - | - | - | - |
| GEN-EX | BCE | 4.57 | 4.20 | 4.83 | 4.96 | 5.29 | 10.12 |
| GEN | BCE | 4.37 | 3.95 | 3.51 | 3.61 | 4.01 | 10.00 |
| GEN-EX | CE | 4.30 | 4.16 | 4.99 | 5.87 | 5.39 | 87.17 |
| GEN | CE | 3.15 | **3.11** | 3.14 | 3.21 | 3.64 | 6.20 |
| GEN-EX | L1 | 4.10 | 4.11 | 4.34 | 4.34 | 4.11 | 5.09 |

**Impact of quantization and decoding functions:** As discussed in Section 3, a real-valued label is quantized to a discrete value in $\{1, 2, ..., N\}$ before applying the encoding function. In Table 14, we show the effect of increasing the number of quantization levels ($N$) on the error for correlation-based decoding ($\mathcal{D}^{\text{GEN}}$, which returns a quantized prediction) and expected correlation-based decoding ($\mathcal{D}^{\text{GEN-EX}}$, which returns a continuous prediction). As shown in the table, there exists a tradeoff between reducing quantization error and using fewer classifiers. The error is lower for 128 quantization levels than it is for 256 as the improvement resulting from fewer binary classifiers is higher than the increase in quantization error. Moreover, the use of proposed decoding function $\mathcal{D}^{\text{GEN-EX}}$ for regression consistently results in lower error compared to $\mathcal{D}^{\text{GEN}}$.

Table 14: Impact of the quantization and decoding functions on NME for facial landmark detection.

|  | COFW | | | 300W | | |
|---|---|---|---|---|---|---|
| Quantization levels | 64 | 128 | 256 | 64 | 128 | 256 |
| $\mathcal{E}^{\text{BEL-U}} + \mathcal{D}^{\text{GEN}}$ | 3.66 | 3.51 | 3.46 | 3.79 | 3.59 | 3.46 |
| $\mathcal{E}^{\text{BEL-U}} + \mathcal{D}^{\text{GEN-EX}}$ | 3.46 | 3.41 | 3.44 | 3.54 | 3.47 | 3.44 |
| $\mathcal{E}^{\text{BEL-J}} + \mathcal{D}^{\text{GEN}}$ | 3.65 | 3.49 | 3.43 | 3.76 | 3.58 | 3.46 |
| $\mathcal{E}^{\text{BEL-J}} + \mathcal{D}^{\text{GEN-EX}}$ | 3.45 | **3.40** | 3.42 | 3.52 | 3.45 | **3.43** |

**Impact of the number of training samples on BEL:** As discussed in Section 4, the performance of different encoding functions varies depending on the availability of sufficient training data. In Table 15, we analyze the effect of the number of available training samples for both simple and complex encodings. We use the number of bit transitions as a measure of the complexity of a classifier. As the number of training samples decreases, simpler encodings (U and J) perform better than more complex encodings (B1JDJ, B2JDJ, and HEXJ). Using a more complex encoding reduces the number of classifiers; however, it increases each classifier's complexity (i.e. the number of bit transitions) and thus performs poorly with less training data.

Table 15: Effect of training dataset size on optimal encoding function for facial landmark detection. BCE loss function and GEN-EX decoding function are used for the training and evaluation.

| Encoding | #Classifiers/label | #bit transitions/classifier | Reduction in the number training samples | | | | | | |
|---|---|---|---|---|---|---|---|---|---|
|  |  |  | 0% | 20% | 40% | 60% | 80% | 90% | 95% |
| COFW (FLD1) | | | | | | | | | |
| U | 256 | 1 | 3.45 | 3.48 | 3.55 | 3.72 | 3.94 | 4.52 | 6.29 |
| J | 128 | 2 | 3.43 | 3.48 | 3.51 | 3.61 | **3.88** | **4.32** | **5.39** |
| B1JDJ | 65 | 4 | 3.42 | **3.44** | 3.52 | **3.60** | 4.11 | 4.50 | 5.68 |
| B2JDJ | 34 | 8 | **3.41** | 3.45 | **3.48** | 3.80 | 3.94 | 4.80 | 6.56 |
| HEXJ | 17 | 32 | 3.47 | 3.69 | 3.78 | 4.03 | 4.61 | 5.48 | 6.69 |
| 300W (FLD2) | | | | | | | | | |
| U | 256 | 1 | 3.48 | 3.55 | 3.58 | 3.64 | 3.89 | 4.26 | 5.66 |
| J | 128 | 2 | 3.46 | 3.56 | 3.52 | 3.58 | **3.79** | **4.04** | **4.58** |
| B1JDJ | 65 | 4 | 3.43 | 3.48 | 3.53 | 3.61 | 3.89 | 4.31 | 6.10 |
| B2JDJ | 34 | 8 | 3.42 | **3.47** | **3.51** | **3.54** | 3.88 | 4.50 | 5.80 |
| HEXJ | 17 | 32 | **3.38** | 3.64 | 3.73 | 3.97 | 4.41 | 5.38 | 6.60 |
| WFLW (FLD3) | | | | | | | | | |
| U | 256 | 1 | 4.60 | 4.67 | 4.83 | 5.00 | 5.37 | 6.04 | 7.46 |
| J | 128 | 2 | 4.51 | 4.60 | 4.65 | 4.84 | 5.23 | **5.64** | **6.39** |
| B1JDJ | 65 | 4 | 4.43 | **4.44** | 4.52 | 4.66 | 5.08 | 5.90 | 8.39 |
| B2JDJ | 34 | 8 | 4.38 | 4.46 | **4.49** | **4.61** | **5.02** | 5.95 | 8.78 |
| HEXJ | 17 | 32 | **4.37** | 4.60 | 4.72 | 4.96 | 5.72 | 6.86 | 8.09 |
| AFLW (FLD4) | | | | | | | | | |
| U | 256 | 1 | 1.50 | 1.53 | 1.53 | 1.56 | 1.61 | 1.68 | 1.83 |
| J | 128 | 2 | 1.50 | 1.51 | 1.52 | 1.54 | 1.60 | 1.68 | 1.79 |
| B1JDJ | 65 | 4 | **1.47** | **1.50** | 1.52 | 1.54 | 1.60 | 1.67 | 1.78 |
| B2JDJ | 34 | 8 | **1.47** | **1.50** | **1.50** | **1.52** | 1.57 | **1.64** | **1.73** |
| HEXJ | 17 | 32 | 1.49 | 1.54 | 1.54 | 1.55 | 1.59 | 1.71 | 1.89 |

**Impact of reflected binary conversion:** As mentioned in Section 3.2, we use reflected binary to increase the distance between distant labels based on the design properties of suitable regression encodings we proposed. Table 16 shows the impact of using reflected binary conversion on error for

facial landmark detection benchmarks. As shown in the table, the use of reflected binary significantly reduces the error.

Table 16: Effect of reflected binary conversion for B1JDJ encoding on facial landmark detection. Here, BCE loss and GEN-EX decoding functions are used.

| | COFW | 300W | WFLW | AFLW |
|---|---|---|---|---|
| B1JDJ | 3.43 | 3.46 | 4.43 | 1.47 |
| B1JDJ- w/o reflected binary | 4.13 | 4.43 | 5.70 | 1.97 |

**Use of binary heamaps:** Facial landmark detection approaches typically use heatmap regression. We also evaluate BEL-H-x, in which the real-valued heatmaps are converted to binary heatmaps with 8 quantization levels. Table 17 shows the impact of using binary heatmaps on error for facial landmark detection benchmarks. For unary code, a $64 \times 64$ real-valued heatmap of one facial landmark is converted to eight $64 \times 64$ binary heatmaps, resulting in $32,768$ ($8 \times 64 \times 64$) binary classifiers compared to $512$ for BEL-U. We believe that training a high number of binary classifiers results in high error for BEL-H-x.

Table 17: Comparison of BEL with heatmaps for facial landmark detection. Here, BCE loss and GEN-EX decoding functions are used.

| | FLD1 | FLD2 | FLD3 | FLD4 |
|---|---|---|---|---|
| BEL-U | 3.45 | 3.46 | 4.60 | 1.50 |
| BEL-J | 3.48 | 3.46 | 4.51 | 1.50 |
| BEL-H-U | 4.13 | 4.43 | 5.70 | 1.97 |
| BEL-H-J | 10.17 | 33.02 | 22.50 | 2.99 |

**Hyperparameter $\theta$:** As shown in Figure 4b, we introduce a feature vector of size $\theta$ before the output layer. Figure 6 compares the decrease in the error for different encodings and $\theta$ values. We observe that more complex encodings benefit more from an increase in the value of $\theta$, while a lower value of $\theta$ can be used for simpler encodings.

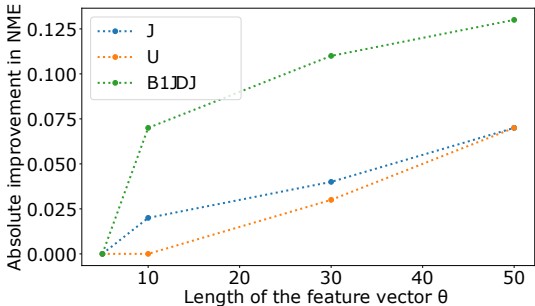

Figure 6: Effect of $\theta$ on error for different encodings on FLD1.

**Impact of increasing the number of fully connected layers:** For BEL, we propose to add a fully connected bottleneck layer in the regressor to reduce the feature vector size to $\theta$ and thus decrease the number of parameters in the regressor. We perform an ablation study to study the impact of this added fully connected layer on relative performance of direct regression, multiclass classification, and binary encoded labels. Table 18 provides the error (MAE or NME) for direct regression and multiclass classification with one or two fully connected layers after the feature extractor. Further, we evaluate BEL, direct regression, and multiclass classification for higher number of fully connected layers as shown in Table 19. We observe that increasing the number of fully connected layers in direct regression and multiclass classification does not improve the accuracy for most benchmarks (possibly due to overparameterization). BEL with two fully connected layers outperforms direct regression and multiclass classification in both cases. Furthermore, even for a higher number of fully connected

Table 18: Impact increasing number of fully connected layers in direct regression and multiclass classification on the error (MAE or NME).

| Benchmark | Direct regression | | Multiclass classification | | BEL |
|---|---|---|---|---|---|
| | 1 FC layer | 2 FC layers | 1 FC layer | 2 FC layers | 2 FC layers |
| HPE1 | 4.76 | 5.19 | 4.49 | 4.82 | **3.37** |
| HPE2 | 5.65 | 5.59 | 5.31 | 5.42 | **4.77** |
| HPE3 | 3.40 | 3.54 | 4.45 | 4.54 | **3.12** |
| HPE4 | 4.14 | 4.22 | 5.14 | 5.45 | **3.90** |
| FLD1 | 3.60 | 3.63 | 3.58 | 3.56 | **3.34** |
| FLD2 | 3.54 | 3.58 | 3.51 | 3.62 | **3.36** |
| FLD3 | 4.64 | 4.63 | 4.50 | 4.64 | **4.33** |
| FLD4 | 1.51 | 1.51 | 1.56 | 1.53 | **1.47** |
| AE1 | 2.44 | 2.35 | 2.75 | 2.81 | **2.27** |
| AE2 | 3.21 | 3.14 | 3.38 | 3.40 | **3.11** |
| PN | 4.24 | 4.33 | 4.56 | 5.74 | **3.11** |

layers in BEL, the suitability of an encoding function varies with the dataset, demonstrating the importance of BEL design space.

Table 19: Impact increasing number of fully connected layers in direct regression, multiclass classification, and BEL. GEN-EX decoding function and BCE loss function are used for BEL.

| Benchmark | # FC layers (size of FC layers) | Direct regression | Multiclass classification | U | J | B1JDJ | B2JDJ | HEXJ |
|---|---|---|---|---|---|---|---|---|
| | 1 (1024-x) | 3.6 | 3.58 | - | - | - | - | - |
| FLD1 | 2 (1024-30-x) | 3.63 | 3.56 | 3.45 | 3.43 | 3.42 | **3.41** | 3.47 |
| | 3 (1024-30-10-x) | 3.63 | 3.94 | 3.55 | **3.47** | 3.82 | 4.02 | 3.62 |
| | 1 (1024-x) | 3.54 | 3.51 | - | - | - | - | - |
| FLD2 | 2 (1024-10-x) | 3.58 | 3.62 | 3.48 | 3.46 | 3.43 | 3.42 | **3.38** |
| | 3 (1024-30-10-x) | 3.55 | 3.78 | **3.42** | 3.46 | 3.5 | 3.61 | 3.52 |

**Training-validation set based evaluation:** Ideally, a validation set should be used for model selection. Hence we have reevaluated the benchmarks with a validation set to select the best design parameters and the best model (i.e., which model is the best over multiple epochs). Since datasets used in benchmarks do not provide separate validation datasets, we use $20\%$ of the training data as a validation set. Since earlier works use $100\%$ training data for the reported results and use test error for model selection, we have re-run specialized approaches (if possible), direct regression, and multiclass classification. It was not possible for us to re-run experiments for all specialized approaches due to resource constraints, and the comparison is conservative for many benchmarks.

Table 20 compares different regression approaches for this evaluation setup. Note that the additional results do not diminish the effectiveness of BEL and BEL outperforms direct regression and multiclass classification for all benchmarks and specialized approaches for several benchmarks.

## B  EXPECTED ERROR DERIVATION

This section explains the expected error equations used to compare BEL-U and BEL-J in Section 3.1. We first explain the encoding and decoding function used for BEL-U and derive the relation between the expected regression error and classification error for BEL-U. Then, we explain the encoding/decoding functions and expected error relation for BEL-J.

### B.1  PRELIMINARIES

Given a sample $i$ drawn from a dataset with minimum label $a$ and maximum label $b$, let $y_i \in [a, b]$ represent the target label for that sample. Assuming uniform quantization, the range of target labels can be quantized using $q : [a, b] \rightarrow \{1, 2, ..., N\}$ through Equation 7.

$$q(y_i) = (y_i - a) * \frac{N - 1}{b - a} + 1 \tag{7}$$

Table 20: Comparison of BEL with different regression approaches. Specialized approaches for each benchmark are described in Table 1

| | Error (MAE or NME) / Model size | | | |
|---|---|---|---|---|
| Approach (% training set) | HPE1 | HPE2 | HPE3 | HPE4 |
| Specialized approach (100%) | - | - | **3.40** / 69.8M | 4.14 / 69.8M |
| Specialized approach (80%) | - | - | 4.08±0.11 / 69.8M | 4.69±0.02 / 69.8M |
| Direct regression (80%) | 6.12±0.02 / 23.5M | 5.97±0.09 / 23.5M | 4.08±0.11 / 69.8M | 4.67+4.70 / 69.8M |
| Multiclass classification (80%) | 5.38±0.03 / 24.2M | 5.60±0.13 / 24.8M | 5.58±0.04 / 72.0M | 5.86±0.10 / 72.0M |
| BEL (80%) | **3.91**±0.08 / 23.6M | **4.91**±0.10 / 23.6M | 3.50±0.08 / 69.8M | **3.99**±0.04 / 69.8M |
| BEL $\mathcal{E}/\mathcal{D}/\mathcal{L}$ functions | U/GEN-EX/L2 | U/GEN-EX/BCE | B1JDJ/GEN-EX/BCE | U/GEN-EX/BCE |

| Approach (%training set) | FLD1 | FLD2 | FLD3 | FLD4 |
|---|---|---|---|---|
| Specialized approach (100%) | 3.45 / 9.6M | **3.32** / 9.6M | **4.32** / 9.6M | 1.57 / 9.6M |
| Direct regression (80%) | 3.70±0.04 / 10.2M | 3.69±0.06 / 10.2M | 4.71±0.02 / 10.2M | 1.51±0.01 / 10.2M |
| Multiclass classification (80%) | 3.64±0.02 / 25.4M | 3.68±0.02 / 45.2M | 4.77±0.02 / 61.3M | 1.56 ±0.01 / 20.1M |
| BEL (80%) | **3.35**±0.02 / 10.6M | 3.40±0.03 / 11.2M | 4.37±0.01 / 11.7M | **1.48**±0.01 / 10.8M |
| BEL $\mathcal{E}/\mathcal{D}/\mathcal{L}$ functions | HEXJ/GEN-EX/CE | U/GEN-EX/CE | B1JDJ/GEN-EX/CE | B1JDJ/GEN-EX/CE |

| Approach (% training set) | AE1 | AE2 | PN | |
|---|---|---|---|---|
| Specialized approach (100%) | 2.49 / 21.3M | 3.47 / 21.3M | 4.24 / 1.8M | |
| Direct regression (80%) | 2.45 ±0.01 / 23.1M | 3.34 ±0.02 / 23.1M | 4.56 ±0.45 / 1.8M | |
| Multiclass classification (80%) | 2.85 ±0.03 / 23.1M | 3.47 ±0.05 / 23.1M | 6.37 ±0.00 / 1.9M | |
| BEL (80%) | **2.36** ±0.01 / 23.1M | **3.20** ±0.00 / 23.1M | **3.49** ±0.01 / 1.8M | |
| BEL $\mathcal{E}/\mathcal{D}/\mathcal{L}$ functions | J/BEL-J/BCE | B1JDJ/GEN-EX/L1 | J/GEN/CE | |

We define the encoding function $\mathcal{E} : \{1, 2, ..., N - 1\} \to \{0, 1\}^M$ to convert a target quantized level $Q_i \in \{1, 2, ..., N - 1\}$ to a binary code $B_i \in \{0, 1\}^M$. We further define the decoding function $\mathcal{D} : \{0, 1\}^M \to [a, b]$ to convert the predicted binary code $\hat{B}_i$ to the predicted label $\hat{y}_i$.

Although the decoding functions used in this analysis predict the quantized label and introduce quantization error, we do not include quantization error in the expected absolute error for our analysis as it is constant for both BEL-U and BEL-J. The expected value of absolute error between the target $y$ and predicted labels $\hat{y}$ is used for the analysis as typically mean absolute error is used as the evaluation metric in regression problems.

Let us denote the error probability of a binary classifier $C^k$ used to predict bit $k$ in a binary code $B_i = \mathcal{E}(n)$ as $e_k(n)$, where $n$ is the target quantized label $Q_i$. Then,

$$
\begin{aligned}
e_k(n) &= \mathbb{E}(|\hat{b}_i^k - b_i^k|) \\
&= Pr(\hat{b}_i^k = F) \text{ for target label } Q_i = n \text{ and target binary code } B_i = \mathcal{E}(n)
\end{aligned}
\tag{8}
$$

where $\hat{b}_i^k = T$ indicates a correct binary classification by classifier $C^k$ ($\hat{b}_i^k == b_i^k$) for sample $i$ and $\hat{b}_i^k = F$ indicates an incorrect binary classification by classifier $C^k$ ($\hat{b}_i^k \neq b_i^k$) for sample $i$.

## B.2 EXPECTED ERROR FOR BEL-U

**Encoding and decoding functions:** The encoding and decoding functions for BEL-U are defined as:

$$
\mathcal{E}^{\text{BEL-U}}(Q_i) = b_i^1, b_i^2, .., b_i^{N-2}, \text{ where } b_i^k = \begin{cases} 1, & k < Q_i \\ 0, & \text{Otherwise} \end{cases}
\tag{9}
$$

$$
\mathcal{D}^{\text{BEL-U}}(\hat{b}_i^1, \hat{b}_i^2, ..., \hat{b}_i^{N-2}) = \sum_{k=1}^{N-2} \hat{b}_i^k + 1
\tag{10}
$$

**Expected error:** For target quantized label $Q_i = n$ ($n \in \{1, 2, ..., N-1\}$), ignoring the quantization error, the expected error between target $y_i$ and predicted label $\hat{y}_i$ can be derived as:

$$
\begin{aligned}
\mathbb{E}(|\hat{y}_i^{\text{BEL-U}} - y_i|) &= \mathbb{E}\left(\left| \sum_{k=1}^{N-2} \left(\hat{b}_i^k + 1\right) - \sum_{k=1}^{N-2} \left(b_i^k + 1\right) \right|\right) \\
&= \mathbb{E}\left(\left| \sum_{k=1}^{N-2} (\hat{b}_i^k - b_i^k) \right|\right) \\
&\leqslant \mathbb{E}\left( \sum_{k=1}^{N-2} |\hat{b}_i^k - b_i^k| \right) \\
&= \sum_{k=1}^{N-2} \mathbb{E}|\hat{b}_i^k - b_i^k| \\
&= \sum_{k=1}^{N-2} e_k(n) \text{ (using Equation 8)}
\end{aligned}
\tag{11}
$$

For a uniform distribution of target labels in the range $[1, N-1]$, the expected error can be derived as:

$$
\mathbb{E}(|\hat{y}^{\text{BEL-U}} - y|) \leqslant \frac{1}{N-1} \sum_{n=1}^{N-1} \sum_{k=1}^{N-2} e_k(n) \tag{12}
$$

## B.3 EXPECTED ERROR FOR BEL-J

**Encoding and decoding functions:** For target quantized label $Q_i \in \{1, 2, ..., N-1\}$, BEL-J encoding requires $\frac{N}{2}$ bits/binary classifiers. The encoding for BEL-J can be defined as:

$$
\mathcal{E}^{\text{BEL-J}}(Q_i) = b_i^1, b_i^2, .., b_i^{\frac{N}{2}}, \text{where } b_i^k = \begin{cases} 1, & \frac{N}{2} - Q_i < k \leqslant N - Q_i \\ 0, & \text{Otherwise} \end{cases} \tag{13}
$$

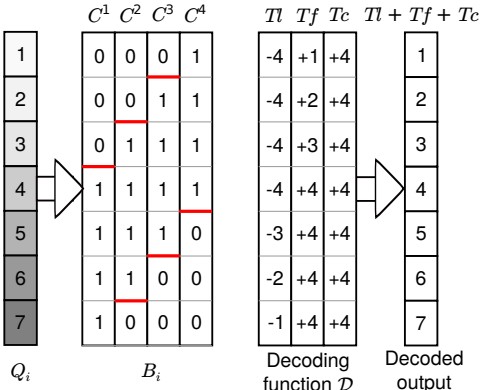

Figure 7: Encoding and Decoding functions' output for BEL-J approach and label $y \in [1, N-1]$, where $N = 8$. Decoding function's output is calculated using $y' = Tl + Tf + Tc$, where $Tl = -\max_{k \in \{1...\frac{N}{2}\}} k\hat{b}_i^k$, $Tf = \max_{k \in \{1...\frac{N}{2}\}} \left(\frac{N}{2} - k + 1\right)\hat{b}_i^k$, and $Tc = \frac{N}{2}$.

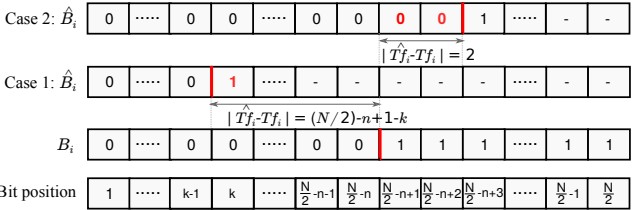

Figure 8: Effect of classifier error on $\hat{T}f_i - Tf_i$ for label $Q_i = n$. Case 1 and case 2 represent erroneous outputs. 0/1 highlighted in red color represents an error in the classifier's output. "-" represents error/no error in both cases.

Similarly, the decoding functions for BEL-J can be defined as:

$$\mathcal{D}^{\text{BEL-J}}(\hat{B}_i) = Tl(\hat{B}_i) + Tf(\hat{B}_i) + Tc$$

$$\text{where, } Tl(\hat{B}_i) = -\max_{k \in \{1...\frac{N}{2}\}} k\hat{b}_i^k$$

$$Tf(\hat{B}_i) = \max_{k \in \{1...\frac{N}{2}\}} \left(\frac{N}{2} - k + 1\right)\hat{b}_i^k, Tc = \frac{N}{2}$$

(14)

In Equation 14, $Tl()$ finds the location of the last occurrence of "1" in the predicted binary code $\hat{B}_i$. Similarly, $Tf()$ finds the location of the first occurrence of "1" in the binary code $\hat{B}_i$. Figure 7 gives examples of binary codes for label $Q_i \in \{1, 2, ..., 7\}$ and the corresponding values of the different terms in Equation 14. For example, for label $Q_i = 3$, the binary code is "0111". Here, the last occurrence of "1" is at position 4, and $Tl = -4$. Similarly, the first occurrence of "1" is at position 2, and $Tf = (4 + 1) - 2 = 3$.

**Expected error:** For BEL-J code, binary classifiers $(C^1, C^2, ..., C^{\frac{N}{2}})$ are used. For a given input sample i, an error in any of the binary classifiers' outputs $(\hat{b}_i^1, \hat{b}_i^2, ..., \hat{b}_i^{\frac{N}{2}})$ will result in an error between $Tf(\hat{B}_i)/Tl(\hat{B}_i)$ and $Tf(B_i)/Tl(B_i)$ in Equation 14. We refer to $Tf(\hat{B}_i)$ and $Tl(\hat{B}_i)$ as $\hat{T}f_i$ and $\hat{T}l_i$ (predicted binary code), and $Tf(B_i)$ and $Tl(B_i)$ as $Tf_i$ and $Tl_i$ (target binary code) for brevity. Expected value of the absolute error can be further expanded as:

$$\mathbb{E}(|\hat{y}_i^{\text{BEL-J}} - y_i|) = \mathbb{E}(|\hat{T}f_i + \hat{T}l_i + Tc - (Tf_i + Tl_i + Tc)|)$$
$$= \mathbb{E}(|(\hat{T}f_i - Tf_i) + (\hat{T}l_i - Tl_i)|)$$
$$\leqslant \mathbb{E}(|\hat{T}f_i - Tf_i| + |\hat{T}l_i - Tl_i|)$$
$$= \mathbb{E}(|\hat{T}f_i - Tf_i|) + \mathbb{E}(|\hat{T}l_i - Tl_i|)$$

(15)

Thus, the sum of the expected error of $Tf()$ and $Tl()$ is the upper bound of the label's expected error. Further, we derive the relation between binary classifiers' error probabilities and $\mathbb{E}(|\hat{T}f_i - Tf_i|)$ and $\mathbb{E}(|\hat{T}l_i - Tl_i|)$.

We consider $Q_i = n$, where $1 \leqslant n \leqslant \frac{N}{2}$ for our derivation. In such a case, $Tf_i = n$ and $Tl_i = -\frac{N}{2}$. However, as the code is symmetric around $Q_i = \frac{N}{2}$, it can be shown that the derived equation for $\mathbb{E}|\hat{y}_i - y_i|$ can be used for $1 \leqslant Q_i \leqslant N - 1$.

**1. Derivation of $\mathbb{E}|\hat{T}f_i - Tf_i|$:** As shown in Equation 14, $Tf()$ finds the location $k$ of the first occurrence of "1" in the binary sequence. In the case of an erroneous binary sequence, the position of the first occurrence of "1" might shift, which results in an error between $\hat{T}f_i$ and $Tf_i$. Figure 8 shows examples of the correct and erroneous outputs of classifiers for label $Q_i = n$. For label $Q_i = n$, $b_i^k = 0$ for $k \in \{1, 2, ..., \frac{N}{2} - n\}$ and $b_i^k = 1$ for $k \in \{\frac{N}{2} - n + 1, \frac{N}{2} - n + 2, ..., \frac{N}{2}\}$.

For case 1, error in a classifier $C^k, k \in \{1, 2, ..., \frac{N}{2} - n\}$ is considered, where $b_i^k = 0$ and $\hat{b}_i^k = 1$. For $k \in \{1, 2, ..., \frac{N}{2} - n\}$, an error at classifier $C^k$ will result in erroneous $\hat{T}f_i$ only if all proceeding classifiers are correct, since if any of the proceeding classifier $z$ is incorrect, i.e. $\hat{b}_i^z = 1$, then the location of the first occurrence of "1" will be shifted to $z$, and any error in the following classifiers

will not affect the value of $\hat{T}f$. Such a case $(\hat{b}_i^1 = T, \hat{b}_i^2 = T, ..., \hat{b}_i^{k-1} = T, \hat{b}_i^k = F, \hat{b}_i^{k+1} = T/F, ..., \hat{b}_i^{\frac{N}{2}} = T/F)$ considers a total of $2^{\frac{N}{2}-k}$ combinations out of $2^{\frac{N}{2}}$ for $k \in \{1, 2, ..., \frac{N}{2} - n\}$. Assuming that the binary classifiers are mutually independent, the error value and the probability of this combination can be shown to be:

$$|\hat{T}f_i - Tf_i| = \left(\frac{N}{2} - n + 1 - k\right) \tag{16}$$

$$Pr(\hat{b}_i^1 = T, \hat{b}_i^2 = T, ..., \hat{b}_i^{k-1} = T, \hat{b}_i^k = F) = Pr(\hat{b}^1 = T)Pr(\hat{b}_i^2 = T)...Pr(\hat{b}_i^{k-1} = T)Pr(\hat{b}^k = F)$$

$$= \left(\prod_{j=1}^{k-1}(1 - e_j(n))\right) \cdot e_k(n) \tag{17}$$

The above term considers combinations $(b'^1 = T, b'^2 = T, ..., b'^{k-1} = T, b'^k = F, b'^{k+1} = T/F, ..., b'^{\frac{N}{2}=T/F})$ for $k \in \{1, 2, ..., \frac{N}{2} - n\}$, which constitutes to a total of $\sum_{k=1}^{\frac{N}{2}-n} 2^{\frac{N}{2}-k}$ combinations out of $2^{\frac{N}{2}}$.

For case 2, error in a classifier $C^k, k \in \{\frac{N}{2} - n + 1, \frac{N}{2} - n + 2, ..., \frac{N}{2}\}$ is considered, where $b_i^k = 1$ and $\hat{b}_i^k = 0$. We consider a combination $(\hat{b}_i^1 = T, \hat{b}_i^2 = T, ..., \hat{b}_i^{\frac{N}{2}-n} = T, \hat{b}_i^{\frac{N}{2}-n+1} = F, ..., \hat{b}_i^{k-1} = F, \hat{b}_i^k = T, \hat{b}_i^{k+1} = T/F, ..., \hat{b}_i^{\frac{N}{2}=T/F})$. For this case, the position of the first occurrence of "1" will be moved to $k$, which will result in erroneous $\hat{T}f_i$. Such a case would cover $2^{\frac{N}{2}-k}$ combinations out of $2^{\frac{N}{2}}$ for $k \in \{\frac{N}{2} - n + 1, \frac{N}{2} - n + 2, ..., \frac{N}{2}\}$. The error value and the probability of this combination can be shown to be:

$$|\hat{T}f_i - Tf_i| = \left(k - (\frac{N}{2} - n + 1)\right) \tag{18}$$

$$Pr(\hat{b}_i^1 = T, \hat{b}_i^2 = T, ..., \hat{b}_i^{\frac{N}{2}-n} = T, \hat{b}_i^{\frac{N}{2}-n+1} = F, ..., \hat{b}_i^{k-1} = F, \hat{b}_i^k = T) =$$

$$\left(\prod_{j=1}^{\frac{N}{2}-n}(1 - e_j(n))\right) \cdot \left(\prod_{j=\frac{N}{2}-n+1}^{k-1} e_j(n)\right) \cdot \left(1 - e_k(n)\right) \tag{19}$$

The above term considers combinations $(\hat{b}_i^1 = T, \hat{b}_i^2 = T, ..., \hat{b}_i^{\frac{N}{2}-n} = T, \hat{b}_i^{\frac{N}{2}-n+1} = F, ..., \hat{b}_i^{k-1} = F, \hat{b}_i^k = T, \hat{b}_i^{k+1} = T/F, ..., \hat{b}_i^{\frac{N}{2}=T/F})$, which constitutes to a total of $\sum_{k=\frac{N}{2}-n+1}^{\frac{N}{2}} 2^{\frac{N}{2}-k}$ combinations out of $2^{\frac{N}{2}}$ for $k \in \{\frac{N}{2} - n + 1, \frac{N}{2} - n + 2, ..., \frac{N}{2}\}$.

Combining Equation 16 to Equation 19, the expected value of $|\hat{T}f_i - Tf_i|$ can be derived as:

$$\mathbb{E}(|\hat{T}f_i - Tf_i|) = \sum_{k=1}^{\frac{N}{2}-n} \left(\frac{N}{2} - n + 1 - k\right) \cdot \left(\prod_{j=1}^{k-1}(1 - e_j(n))\right) \cdot e_k(n)$$

$$+ \sum_{k=\frac{N}{2}-n+1}^{\frac{N}{2}} \left(k - (\frac{N}{2} - n + 1)\right) \cdot \left(\prod_{j=1}^{\frac{N}{2}-n}(1 - e_j(n))\right) \cdot \left(\prod_{j=\frac{N}{2}-n+1}^{k-1}(e_j(n))\right) \cdot \left(1 - e_k(n)\right)$$

$$= \sum_{k=1}^{\frac{N}{2}-n} \left(\frac{N}{2} - n + 1 - k\right) \cdot e_k(n) \cdot \left(\prod_{j=1}^{k-1}(1 - e_j(n))\right) + \sum_{k=\frac{N}{2}-n+1}^{\frac{N}{2}} \left(\prod_{j=\frac{N}{2}-n+1}^{k} e_j(n)\right) \tag{20}$$

The first term in Equation 20 covers $\sum_{k=1}^{\frac{N}{2}-n} 2^{\frac{N}{2}-k}$ combinations and the second term considers $\sum_{k=\frac{N}{2}-n+1}^{\frac{N}{2}} 2^{\frac{N}{2}-k}$ combinations. Adding one combination where all the classifiers are correct, Equation 20 considers all of the possible combinations $2^{\frac{N}{2}}$ to find expected value of $|\hat{T}f_i - Tf_i|$.

Figure 9: Effect of classifier error on $\hat{T}l_i - Tl_i$ for label $Q_i = n$. Case 1 and case 2 represent erroneous outputs. 0/1 highlighted in red color represents an error in the classifier's output. "-" represents error/no error in both cases.

**2. Derivation of $\mathbb{E}\,|\hat{T}l_i - Tl_i|$:** As shown in Equation 14, $Tl()$ finds the location $k$ of the last occurrence of "1" in the binary sequence. In the case of an erroneous binary sequence, the position of the last occurrence of "1" might shift, which results in an erroneous value of $\hat{T}l_i$. Figure 9 shows examples of correct and erroneous outputs of classifiers for label $Q_i = n$.

For case 1, an error in a classifier $C^k, k \in \{\frac{N}{2} - n + 1, \frac{N}{2} - n + 2, ..., \frac{N}{2}\}$ is considered, where $b_i^k = 1$ and $\hat{b}_i^k = 0$. We consider a combination $(\hat{b}_i^{\frac{N}{2}} = F, \hat{b}_i^{\frac{N}{2}-1} = F, ..., \hat{b}_i^{k+1} = F, \hat{b}_i^k = T, \hat{b}_i^{k-1} = T/F, ..., \hat{b}_i^1 = T/F)$. For this case, position of the last occurrence of "1" will be moved to $k$, which will result in erroneous $\hat{T}l_i$. Such a case would cover $2^{k-1}$ combinations out of $2^{\frac{N}{2}}$. The error value and the probability of this combination can be shown to be:

$$|\hat{T}l_i - Tl_i| = \left(\frac{N}{2} - k\right) \tag{21}$$

$$Pr(\hat{b}_i^{\frac{N}{2}} = F, \hat{b}_i^{\frac{N}{2}-1} = F, ..., \hat{b}_i^{k+1} = F, \hat{b}_i^k = T) = \left(\prod_{j=k+1}^{\frac{N}{2}} e_j(n)\right) \cdot (1 - e_k(n)) \tag{22}$$

The above term considers combinations $(\hat{b}_i^{\frac{N}{2}} = F, \hat{b}_i^{\frac{N}{2}-1} = F, ..., \hat{b}_i^{k+1} = F, \hat{b}_i^k = T, \hat{b}_i^{k-1} = T/F, ..., \hat{b}_i^1 = T/F)$ for $k \in \{\frac{N}{2} - n + 1, \frac{N}{2} - n + 2, ..., \frac{N}{2}\}$, which constitutes to a total of $\sum_{k=\frac{N}{2}-n+1}^{\frac{N}{2}} 2^{k-1}$ combinations out of $2^{\frac{N}{2}}$.

For case 2, an error in a classifier $C^k, k \in \{1, 2, ..., \frac{N}{2} - n\}$ is considered, where $b_i^k = 0$ and $\hat{b}_i^k = 1$. We consider a combination $(\hat{b}_i^{\frac{N}{2}} = F, ..., \hat{b}_i^{\frac{N}{2}-n+1} = F, \hat{b}_i^{\frac{N}{2}-n} = T, ..., \hat{b}_i^{k+1} = T, \hat{b}_i^k = F, \hat{b}_i^{k-1} = T/F, ..., \hat{b}_i^1 = T/F)$. For this case, position of the last occurrence of "1" will be moved to $k$, which will result in erroneous $\hat{T}l_i$. Such a case would cover $2^{k-1}$ combinations out of $2^{\frac{N}{2}}$. The error value and the probability of this combination can be shown to be:

$$|\hat{T}l_i - Tl_i| = \left(\frac{N}{2} - k\right) \tag{23}$$

$$Pr(\hat{b}_i^{\frac{N}{2}} = F, ..., \hat{b}_i^{\frac{N}{2}-n+1} = F, \hat{b}_i^{\frac{N}{2}-n} = T, ..., \hat{b}_i^{k+1} = T, \hat{b}_i^k = F) =$$
$$\left(\prod_{j=\frac{N}{2}-n+1}^{\frac{N}{2}} e_j(n)\right) \cdot \left(\prod_{j=k+1}^{\frac{N}{2}-n} (1 - e_j(n))\right) \cdot (e_k(n)) \tag{24}$$

The above term considers combinations $(\hat{b}_i^{\frac{N}{2}} = F, ..., \hat{b}_i^{\frac{N}{2}-n+1} = F, \hat{b}_i^{\frac{N}{2}-n} = T, ..., \hat{b}_i^{k+1} = T, \hat{b}_i^k = F, \hat{b}_i^{k-1} = T/F, ..., \hat{b}_i^1 = T/F)$ for $k \in \{1, 2, ..., \frac{N}{2} - n\}$, which constitutes to a total of $\sum_{k=1}^{\frac{N}{2}-n} 2^{k-1}$ combinations out of $2^{\frac{N}{2}}$.

Combining Equation 21 to Equation 24, the expected value of $|\hat{T}l_i - Tl_i|$ can be derived

as:

$$\mathbb{E}(|\hat{T}l_i - Tl_i|) = \sum_{k=\frac{N}{2}-n+1}^{\frac{N}{2}} \left(\frac{N}{2}-k\right) \cdot \left(\prod_{j=k+1}^{\frac{N}{2}} e_j(n)\right) \cdot (1 - e_k(n))$$

$$+ \sum_{k=1}^{\frac{N}{2}-n} \left(\frac{N}{2}-k\right) \cdot \left(\prod_{j=\frac{N}{2}-n+1}^{\frac{N}{2}} e_j(n)\right) \cdot \left(\prod_{j=k+1}^{\frac{N}{2}-n} (1 - e_j(n))\right) \cdot (e_k(n)) \quad (25)$$

$$= \sum_{k=\frac{N}{2}-n+1}^{\frac{N}{2}} \left(\prod_{j=k}^{\frac{N}{2}} e_j(n)\right) + \left(\prod_{j=\frac{N}{2}-n+1}^{\frac{N}{2}} e_j(n)\right) \cdot \sum_{k=1}^{\frac{N}{2}-n} \left(\prod_{j=k}^{\frac{N}{2}-n} (1 - e_j(n))\right)$$

The first term in Equation 25 covers $\sum_{k=\frac{N}{2}-n+1}^{\frac{N}{2}} 2^{k-1}$ combinations and the second term considers $\sum_{k=1}^{\frac{N}{2}-n} 2^{k-1}$ combinations. Adding one combination where all the classifiers are correct, Equation 25 considers all of the possible combinations $2^{\frac{N}{2}}$ to find expected value of $|\hat{T}l_i - Tl_i|$.

Combining Equation 15, Equation 20, and Equation 25, the expected value of error for $Q_i = n$ in terms of classifiers' error probabilities can be derived as:

$$\mathbb{E}(\hat{y}_i^{\text{BEL-J}} - y_i) \leqslant \sum_{k=1}^{\frac{N}{2}-n} \left(\frac{N}{2}-n+1-k\right) \cdot e_k(n) \cdot \left(\prod_{j=1}^{k-1} (1-e_j(n))\right) + \sum_{k=\frac{N}{2}-n+1}^{\frac{N}{2}} \left(\prod_{j=\frac{N}{2}-n+1}^{k} e_j(n)\right)$$

$$+ \sum_{k=\frac{N}{2}-n+1}^{\frac{N}{2}} \left(\prod_{j=k}^{\frac{N}{2}} e_j(n)\right) + \left(\prod_{j=\frac{N}{2}-n+1}^{\frac{N}{2}} e_j(n)\right) \cdot \sum_{k=1}^{\frac{N}{2}-n} \left(\prod_{j=k}^{\frac{N}{2}-n} (1 - e_j(n))\right) \quad (26)$$

As the binary code is symmetric around $\frac{N}{2}$ as shown in Figure 7, the expected errors for label $y_i \in [1, \frac{N}{2}]$ can be mirrored to find expected errors for label $y_i \in [\frac{N}{2}, N-1]$. For a uniform distribution of target labels in the range $[1, N-1]$, the expected error can be derived as:

$$\mathbb{E}(\hat{y}^{\text{BEL-J}} - y) \leqslant \frac{1}{N-1} \sum_{n=1}^{N-1} \Big[ \sum_{k=1}^{\frac{N}{2}-n} \left(\frac{N}{2}-n+1-k\right) \cdot e_k(n) \cdot \left(\prod_{j=1}^{k-1} (1-e_j(n))\right) + \sum_{k=\frac{N}{2}-n+1}^{\frac{N}{2}} \left(\prod_{j=\frac{N}{2}-n+1}^{k} e_j(n)\right)$$

$$+ \sum_{k=\frac{N}{2}-n+1}^{\frac{N}{2}} \left(\prod_{j=k}^{\frac{N}{2}} e_j(n)\right) + \left(\prod_{j=\frac{N}{2}-n+1}^{\frac{N}{2}} e_j(n)\right) \cdot \sum_{k=1}^{\frac{N}{2}-n} \left(\prod_{j=k}^{\frac{N}{2}-n} (1 - e_j(n))\right) \Big] \quad (27)$$

We also verify the equation by comparing the expected value of error based on Equation 26 for $Q_i \in \{1, 2, ..., N-1\}$ with the expected error calculated by $100,000$ random samples of binary sequences for the same error probabilities $e_k(n)$. Figure 10 compares the expected error from Equation 26 and measured from statistical samples, and validates error upper bounds calculated using Equation 26 and Equation 27.

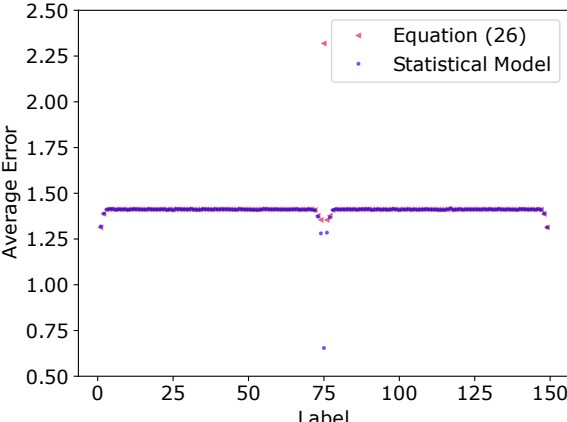

Figure 10: Comparison of expected value of error from Equation 26 and random samples for given error probabilities of the classifiers.

## C    ERROR PROBABILITY OF CLASSIFIERS

It is known that the error/misclassification probability $e_k(n)$ of a classifier tends to increase as the target label value $n$ is closer to the classifier's decision boundaries (Cardoso & Pinto da Costa, 2007). We approximate $e_k(y)$ for a classifier $C^k$ with $t$ bit transitions as a linear combination of $t$ Gaussian distributions. Here, each Gaussian term is centered around a bit transition. Figure 11 shows the empirically observed error probability distributions for different classifiers trained for different combinations of network and dataset. We also show the approximate error probability distribution using a linear combination of Gaussian distributions. Here r is a scalar multiplied with probability density of gaussian distribution and $\sigma$ is the standard deviation (Equation 3 and Equation 4).

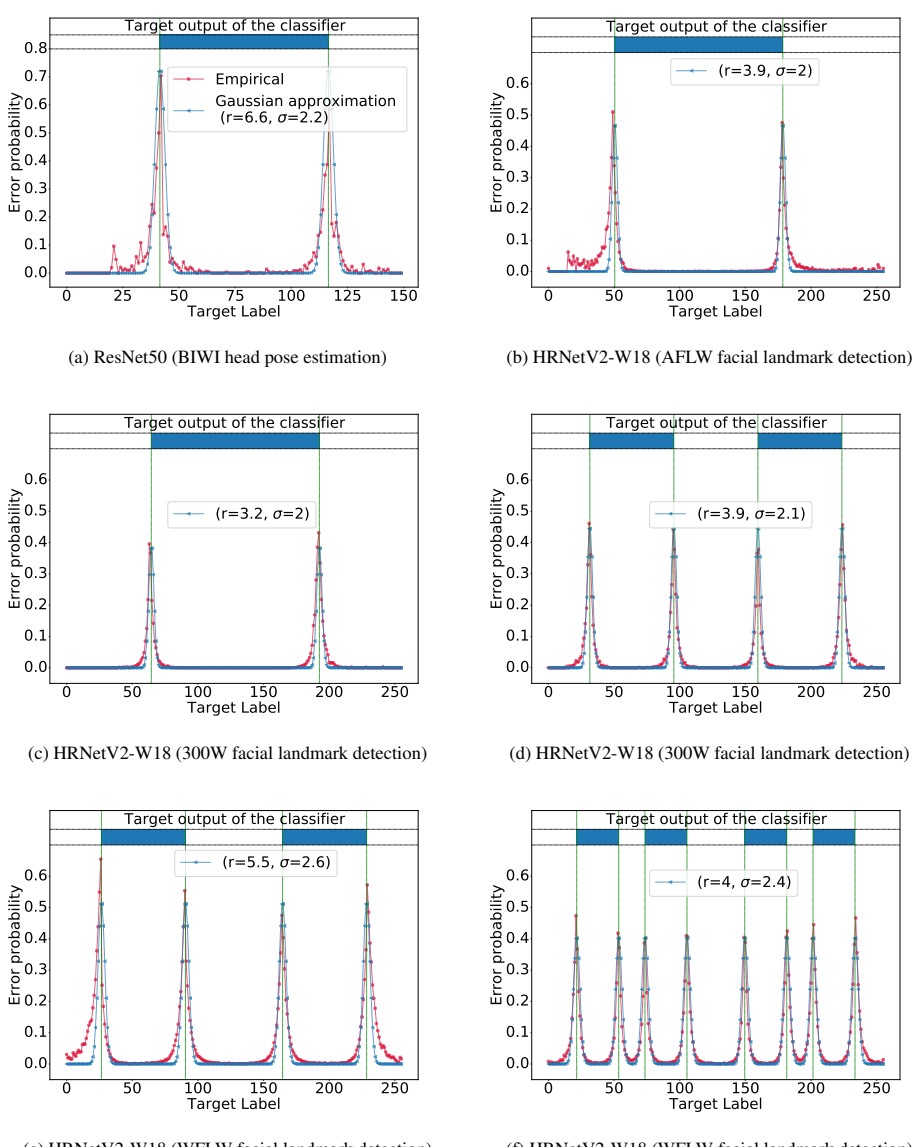

Figure 11: Classification error probability versus target label $y$ for different classifiers. The top horizontal bar represent target output of the classifier. Blue color represents output 1.

# D    EXPERIMENTAL METHODOLOGY

All experiments are conducted on a Linux machine with an Intel i9-9900X processor and an Nvidia RTX 2080 Ti GPU with 11GB of memory. Our code is implemented using Python 3.8.3 with Pytorch 1.5.1 using CUDA 10.2. Our evaluation is averaged over 5 training runs with separate seeds.

## D.1    HEAD POSE ESTIMATION

Head pose estimation aims to find a human head's pose in terms of three angles: yaw, pitch, and roll. In this work, we consider landmark-free 2D head pose estimation.

**Datasets:** We follow the evaluation setting of Hopenet (Ruiz et al., 2018) and FSA-Net (fsa, 2019) and use two evaluation protocols with three widely used datasets: 300W-LP (Zhu et al., 2016), BIWI (Fanelli et al., 2013), and AFLW2000 (Zhu et al., 2016).

Protocol 1: BIWI dataset is used for training and evaluation in this protocol. BIWI dataset consists of 24 videos of 20 subjects with total $15, 128$ frames. Three random splits of 70%-30% images are used for training and evaluation. For the BIWI dataset, the yaw angle is in the range $[-75°, 75°]$, the pitch is in the range $[-65°, 85°]$, and the roll angle is in the range $[-55°, 45°]$.

Protocol 2: In this setting, the synthetic 300W-LP dataset is used for training, consisting of $122, 450$ samples. The trained network is tested on a real-world AFLW2000 dataset. Yaw, pitch, and roll angles are in the range $[-99°, 99°]$ for both datasets.

**Evaluation metrics:** Mean Absolute Error (MAE) between the target and predicted values is used as the evaluation metric for this benchmark. MAE for a regression task is defined as:

$$\text{MAE} = \frac{1}{N} \sum_{i=1}^{N} \frac{1}{P} \sum_{j=1}^{P} |y_{i,j} - \hat{y}_{i,j}| \tag{28}$$

Here, $N$ is the number of test samples, and $P$ is the dimension of the regression task output. For head pose estimation, the dimension of regression output is three (i.e., yaw, pitch, and roll). $y$ is the target, and $\hat{y}$ is the predicted label.

**Network architecture and training parameters:** We evaluate our approach on two models: ResNet-50 and RAFA-Net. With ResNet-50, two runs with different random seeds for each combination of learning rate $\{0.001, 0.0001, 0.00001\}$ and batch size $\{8, 16\}$ are used for hyperparameter tuning. For data augmentation, images are loosely cropped around the center in the training and testing datasets with random flipping. With RAFA-Net, we use the training parameters and data augmentation used in Behera et al. (2021).

We refer to Protocol 1 evaluated with ResNet-50 as **HPE1**, Protocol 1 evaluated with RAFA-Net as **HPE3**, Protocol 2 evaluated with ResNet-50 as **HPE2**, and Protocol 2 evaluated with RAFA-Net as **HPE4**. Table 21 provides a summary of the training parameters used with protocol 1. Table 22 provides a summary of the training parameters used with protocol 2.

Table 21: Training parameters for head pose estimation with protocol 1.

| Approach | Label range/Quantization levels | Optimizer | Epochs | Batch size | Learning rate | Learning rate schedule | Training time (GPU hours) |
|---|---|---|---|---|---|---|---|
| HPE1 | Yaw: $[-75°, 75°]/150$, Pitch:$[-65°, 85°]/150$, Roll: $[-55°, 45°]/100$ | Adam, weight decay=0, momentum = 0 | 50 | 8 | 0.0001 | 1/10 after 30 Epochs | 2 |
| HPE3 | $[-179°, 180°]/360$ | RMSProp, momentum=0, rho = 0.9 | 100 | 16 | 0.001 | - | 6 |

Table 22: Training parameters for head pose estimation with protocol 2.

| Approach | Label range/Quantization levels | Optimizer | Epochs | Batch size | Learning rate | Learning rate schedule | Training time (GPU hours) |
|---|---|---|---|---|---|---|---|
| HPE2 | $[-99°, 99°]/200$ | Adam, weight decay=0, momentum = 0 | 20 | 16 | 0.00001 | 1/10 after 10 Epochs | 4 |
| HPE4 | $[-179°, 180°]/360$ | RMSProp, momentum=0, rho = 0.9 | 100 | 16 | 0.001 | - | 48 |

**Related work** Existing approaches for head pose estimation include stage-wise soft regression (Yang et al., 2018; fsa, 2019), a combination of classification and regression (Mukherjee & Robertson, 2015; Ruiz et al., 2018), and ordinal regression (Hsu et al., 2019). SSR-Net (Yang et al.,

2018) proposes the use of stage-wise soft regression to use the softmax values of classification output to refine the label. FSA-Net (fsa, 2019) proposes extending stage-wise estimation to head pose estimation using feature aggregation. HopeNet (Ruiz et al., 2018) uses a combination of classification and regression loss to train a model for head pose estimation. Whereas, QuatNet (Hsu et al., 2019) proposes a combination of L2 loss and a custom ordinal regression loss. RAFA-Net (Behera et al., 2021) uses an attention based approach for feature extraction with direct regression.

We compare BEL with the performance of related work in Table 23 and Table 24. 95% confidence intervals are given.

Table 23: Landmark-free 2D Head poses estimation evaluation for protocol 1 (HPE1 and HPE3).

| Approach | Feature Extractor | #Params (M) | Yaw | Pitch | Roll | MAE |
|---|---|---|---|---|---|---|
| SSR-Net-MD (Yang et al., 2018) (Soft regression) | SSR-Net | 1.1 | 4.24 | 4.35 | 4.19 | 4.26 |
| FSA-Caps-Fusion (fsa, 2019) (Soft regression) | FSA-Net | 5.1 | _2.89_ | 4.29 | 3.60 | 3.60 |
| Direct regression (L2 loss) | ResNet50 (HPE1) | 23.5 | 4.62 | 5.24 | 4.43 | $4.76 \pm 0.35$ |
| BEL-U/GEN-EX/L2 | ResNet50 (HPE1) | 23.6 | **3.32** | **3.80** | **3.53** | **3.56** $\pm 0.01$ |
| RAFA-Net (Behera et al., 2021) (Direct Regression) | RAFA-Net (HPE3) | 69.8 | 3.07 | 4.30 | _2.82_ | 3.40 |
| BEL-B1JDJ/GEN-EX/BCE | RAFA-Net (HPE3) | 69.8 | **3.21** | _3.34_ | 3.43 | _3.30_ $\pm 0.04$ |

Table 24: Landmark-free 2D Head poses estimation evaluation for protocol 2 (HPE2 and HPE4).

| Approach | Feature Extractor | #Params (M) | Yaw | Pitch | Roll | MAE |
|---|---|---|---|---|---|---|
| SSR-Net-MD (Yang et al., 2018) (Soft regression) | SSR-Net | 1.1 | 5.14 | 7.09 | 5.89 | 6.01 |
| FSA-Caps-Fusion (fsa, 2019) (Soft regression) | FSA-Net | 5.1 | 4.50 | 6.08 | 4.64 | 5.07 |
| HopeNet* ($\alpha = 2$) (Ruiz et al., 2018) (classification + regression loss) | ResNet50 | 23.9 | 6.47 | 6.56 | 5.44 | 6.16 |
| Direct regression (L2 loss) | ResNet50 (HPE2) | 23.5 | 5.85 | 6.34 | 4.80 | $5.65 \pm 0.13$ |
| BEL-U/GEN-EX/BCE | ResNet50 (HPE2) | 23.6 | **4.54** | **5.76** | **3.96** | **4.77** $\pm 0.05$ |
| RAFA-Net (Behera et al., 2021) (Direct Regression) | RAFA-Net (HPE4) | 69.8 | 3.60 | 4.92 | 3.88 | 4.13 |
| BEL-U/GEN-EX/BCE | RAFA-Net (HPE4) | 69.8 | _3.28_ | _4.78_ | _3.55_ | _3.90_ $\pm 0.03$ |

## D.2 FACIAL LANDMARK DETECTION

Facial landmark detection is a problem of detecting the $(x, y)$ coordinates of keypoints in a given face image.

**Datasets** We use the COFW (Burgos-Artizzu et al., 2013), 300W (Sagonas et al., 2013), WFLW (Wu et al., 2018), and AFLW (Köstinger et al., 2011) datasets with data augmentation and evaluation protocols described in (Wang et al., 2020). Data augmentation is performed by random flipping, $0.75 - 1.25$ scaling, and $\pm 30$ degrees in-plane rotation for all the datasets. We use 256 quantization levels for binary-encoded labels.

COFW: The COFW dataset (Burgos-Artizzu et al., 2013) consists of $1,345$ training and $507$ testing images. Each image is annotated with 29 facial landmarks.

300W: This dataset is a combination of HELEN, LFPW, AFW, XM2VTS, and IBUG datasets. Each image is annotated with 68 facial landmarks. The training dataset consists of $3,148$ images. We evaluate the trained model on four test sets: full test set with 689 images, common subset with 554

images from HELEN and LFPW, challenging subset with 135 images from IBUG, and the official test set with 300 indoor and 300 outdoor images.

WFLW: WFLW dataset consists of 7,500 training images where each image is annotated with 98 facial landmarks. Full test dataset consists of 2,500 images. We use test subsets: large pose (326 images), expression (314 images), illumination (698 images), make-up (206 images), occlusion (736 images), and blur (773 images).

AFLW: Each image has 19 annotated facial key points in this dataset. AFLW dataset consists of 20,000 training images where each image is annotated with 19 facial landmarks. The full test dataset consists of 4,836 images, and the frontal test set consists of 1,314 images.

**Evaluation metrics:** Mean Normalized Error (NME) between the target and predicted values is used as the evaluation metric for this benchmark. NME for a regression task is defined as:

$$\text{NME} = \frac{1}{N}\sum_{i=1}^{N}\frac{1}{P}\cdot\frac{1}{L}\sum_{j=1}^{P}|y_{i,j}-\hat{y}_{i,j}|_2 \tag{29}$$

Here, $N$ is the number of test samples, and $P$ is the dimension of the regression task output, i.e., the number of landmarks for facial landmark detection. $y$ is the target, and $\hat{y}$ is the predicted label. $L$ is the normalization factor. . Inter-ocular distance normalization is used for COFW, 300W, and WFLW datasets, and bounding box-based normalization is used for AFLW dataset.

We also report failure rate (f@10%) for some datasets. The failure rate (f@10%) is defined as the fraction of test samples with normalized errors higher than 0.1.

**Network architecture and training parameters:** We evaluate BEL by applying it on HRNetV2-W18. HRNetV2-W18 feature extractor's output is 240 channels of size $64 \times 64$. For heatmap regression, a $1 \times 1$ convolution is used to get $P$ heatmaps of size $64 \times 64$, where $P$ is the number of landmarks. Since BEL-x predicts $(x, y)$ coordinates directly we modify the architecture of HRNetV2-W18 to support direct prediction of landmarks. Figure 12 shows the modified architecture of HRNetV2-W18 for BEL-x.

The state-of-the-art approaches for facial landmark detection uses heatmap regression, which minimizes the pixel-level loss between the predicted and target heatmaps. We evaluate the applicability of BEL on heatmap regression in Appendix A. In contrast, BEL-x predicts $(x, y)$ coordinates directly with 256 quantization levels.

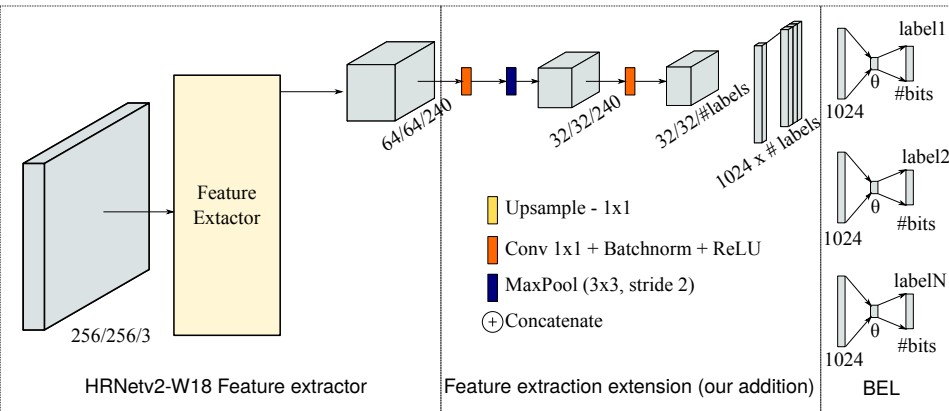

Figure 12: HRNetV2-W18 feature extractor combined with BEL regressor for (x,y) coordinates

We use two runs with different random seeds to decide the learning rate. We consider learning rates $\{0.0003, 0.0005, 0.0007\}$ and $\theta \in \{10, 30\}$.

Table 25 provides a summary of all the training parameters. We refer to HRNetV2-W18 evaluated on COFW as **FLD1**, on 300W as **FLD2**, on WFLW as **FLD3**, and on AFLW as **FLD4**.

Table 25: Training parameters for facial landmark detection for HRNetV2-W18 feature extractor.

| Dataset | Optimizer | Epochs | Batch size | Learning rate (BEL/Direct regression/Multiclass classification) | Learning rate schedule | Training time (GPU hours) |
|---------|-----------|--------|------------|-----------------------------------------------------------------|------------------------|---------------------------|
| COFW | Adam, weight decay=0, momentum = 0 | 60 | 8 | 0.0005/0.0003/ 0.0003 | 1/10 after 30 and 50 Epochs | $\frac{1}{2}$ |
| 300W | Adam, weight decay=0, momentum = 0 | 60 | 8 | 0.0007/0.0003/ 0.0003 | 1/10 after 30 and 50 Epochs | 3 |
| WFLW | Adam, weight decay=0, momentum = 0 | 60 | 8 | 0.0003/0.0003/ 0.0003 | 1/10 after 30 and 50 Epochs | 5 |
| AFLW | Adam, weight decay=0, momentum = 0 | 60 | 8 | 0.0005/0.0005/ 0.0003 | 1/10 after 30 and 50 Epochs | 8 |

**Related work** Facial landmark detection is an extensively studied problem used for facial analysis and modeling. Common regression approaches for this tasks includes regression using MSE loss (Xiong & De la Torre, 2013; Lv et al., 2017), cascaded regression (Miao et al., 2018; Tzimiropoulos, 2015; Zhu et al., 2016; Sun et al., 2013), and coarse-to-fine regression (Sun et al., 2013; Shizhan Zhu et al., 2015; Zhang et al., 2014). State-of-the-art methods for this task learn heatmaps by regression to find facial landmarks. SAN (Dong et al., 2018) augments training data using temporal information and GAN-generated faces. DVLN (Wu & Yang, 2017), CFSS (Shizhan Zhu et al., 2015), LAB (Wu et al., 2018), DSRN (Miao et al., 2018) take advantage of correlations between facial landmarks. DAN (Kowalski et al., 2017) introduces a progressive refinement approach using predicted landmark heatmaps. LAB (Wu et al., 2018) also exploits extra boundary information to improve the accuracy. LUVLi (Kumar et al., 2020) proposes a landmark's location, uncertainty, and visibility likelihood-based loss. Bulat & Tzimiropoulos (2016) proposes the use of binary heatmaps with pixel-wise binary cross-entropy loss. AWing (Wang et al., 2019) proposes adapted wing loss to improve the accuracy of heatmap regression. AnchorFace (Xu et al., 2020) demonstrates that anchoring facial landmarks on templates improves regression performance for large poses. HRNet (Wang et al., 2020) proposes a CNN architecture to maintain high-resolution representations across the network, and uses heatmap regression. The target heatmap is generated by assuming a Gaussian distribution around the landmark location.

We compare BEL with related work in Table 26- 29. 95% confidence intervals are provided.

Table 26: Facial landmark detection results on COFW dataset (FLD1). The failure rate is measured at the threshold 0.1. $\theta = 30$ is used for BEL.

| Approach | Feature Extractor | #Params/ GFlops | Test NME | $FR_{0.1}$ |
|----------|-------------------|-----------------|----------|------------|
| LAB (w B) (Wu et al., 2018) | Hourglass | 25.1/19.1 | 3.92 | 0.39 |
| AWing (Wang et al., 2019)* | Hourglass | 25.1/19.1 | 4.94 | - |
| HRNetV2-W18 (Wang et al., 2020) (Heatmap regression) | HRNetV2-W18 | 9.6/4.6 | 3.45 | 0.19 |
| Direct regression (L2 loss) | HRNetV2-W18 | 10.2/4.7 | $3.96 \pm 0.02$ | 0.29 |
| Direct regression (L1 loss) | HRNetV2-W18 | 10.2/4.7 | $3.60 \pm 0.02$ | 0.29 |
| BEL-HEXJ/GEN-EX/CE | HRNetV2-W18 | 10.6/4.6 | $\mathbf{3.34} \pm 0.02$ | 0.40 |

*Uses different data augmentation for the training

Table 27: Facial landmark detection results on 300W dataset (FLD2). $\theta = 10$ is used for BEL.

| Approach | Feature Extractor | #Params/ GFlops | Test | Common | Challenging | Full |
|---|---|---|---|---|---|---|
| DAN (Kowalski et al., 2017) | - | - | - | 3.19 | 5.24 | 3.59 |
| LAB (w B) (Wu et al., 2018) | Hourglass | 25.1/19.1 | - | 2.98 | 5.19 | 3.49 |
| AnchorFace (Xu et al., 2020) | ShuffleNet-V2 | - | - | 3.12 | 6.19 | 3.72 |
| AWing (Wang et al., 2019)* | Hourglass | 25.1/19.1 | - | 2.72 | 4.52 | 3.07 |
| LUVLi (Kumar et al., 2020) | CU-Net | - | - | 2.76 | 5.16 | 3.23 |
| HRNetV2-W18 (Wang et al., 2020) (Heatmap regression) | HRNetV2-W18 | 9.6/4.6 | - | **2.87** | **5.15** | **3.32** |
| Direct regression (L2 loss) | HRNetV2-W18 | 10.2/4.7 | 4.40 | 3.25 | 5.65 | $3.71 \pm 0.05$ |
| Direct regression (L1 loss) | HRNetV2-W18 | 10.2/4.7 | 4.26 | 3.10 | 5.42 | $3.54 \pm 0.03$ |
| BEL-U/GEN-EX/CE | HRNetV2-W18 | 11.2/4.6 | **4.09** | 2.91 | 5.50 | $3.40 \pm 0.02$ |

∗Uses different data augmentation for the training

Table 28: Facial landmark detection results (NME) on WFLW test (FLD3) and 6 subsets: pose, expression (expr.), illumination (illu.), make-up (mu.), occlusion (occu.) and blur. $\theta = 10$ is used for BEL.

| Approach | Feature Extractor | #Params/ GFlops | Test | Pose | Expr. | Illu. | MU | Occu. | Blur |
|---|---|---|---|---|---|---|---|---|---|
| LAB (w B) (Wu et al., 2018) | Hourglass | 25.1/19.1 | 5.27 | 10.24 | 5.51 | 5.23 | 5.15 | 6.79 | 6.32 |
| AnchorFace (Xu et al., 2020)* | HRNetV2-W18 | -/5.3 | 4.32 | 7.51 | 4.69 | 4.20 | 4.11 | 4.98 | 4.82 |
| AWing (Wang et al., 2019)* | Hourglass | 25.1/19.1 | 4.36 | 7.38 | 4.58 | 4.32 | 4.27 | 5.19 | 4.96 |
| LUVLi (Kumar et al., 2020) | CU-Net | - | 4.37 | - | - | - | - | - | - |
| HRNetV2-W18 (Wang et al., 2020) (Heatmap regression) | HRNetV2-W18 | 9.6/4.6 | 4.60 | 7.94 | 4.85 | 4.55 | 4.29 | 5.44 | 5.42 |
| Direct regression (L2 loss) | HRNetV2-W18 | 10.2/4.7 | $5.56 \pm 0.05$ | 10.17 | 6.13 | 5.49 | 5.29 | 6.83 | 6.52 |
| Direct regression (L1 loss) | HRNetV2-W18 | 10.2/4.7 | $4.64 \pm 0.03$ | 8.13 | 4.96 | 4.49 | 4.45 | 5.41 | 5.25 |
| BEL-B1JDJ/GEN-EX/CE | HRNetV2-W18 | 11.7/4.6 | **$4.36 \pm 0.02$** | **7.53** | **4.64** | **4.28** | **4.19** | **5.19** | **5.05** |

∗Uses different data augmentation for the training

Table 29: Facial landmark detection results on AFLW dataset (FLD4). $\theta = 30$ is used for BEL.

| Approach | Feature Extractor | #Params/ GFlops | Full | Frontal |
|---|---|---|---|---|
| LAB (w/o B) (Wu et al., 2018) | Hourglass | 25.1/19.1 | 1.85 | 1.62 |
| AnchorFace (Xu et al., 2020) | ShuffleNet-V2 | - | 1.56 | |
| LUVLi (Kumar et al., 2020) | CU-Net | - | 1.39 | 1.19 |
| HRNetV2-W18 (Wang et al., 2020) (Heatmap regression) | HRNetV2-W18 | 9.6/4.6 | 1.57 | 1.46 |
| Direct regression (L2 loss) | HRNetV2-W18 | 10.2/4.7 | $2.10 \pm 0.02$ | 1.71 |
| Direct regression (L1 loss) | HRNetV2-W18 | 10.2/4.7 | $1.51 \pm 0.01$ | 1.34 |
| BEL-B1JDJ/GEN-EX/CE | HRNetV2-W18 | 10.8/4.6 | **$1.47 \pm 0.00$** | **1.30** |

∗Uses different data augmentation for the training

## D.3 AGE ESTIMATION

Age estimation aims to predict the age given an image of a human head.

**Datasets**    We use the MORPH-II (Ricanek & Tesafaye, 2006) and AFAD (Niu et al., 2016) datasets for our evaluation. Cumulative Score (CS) and MAE are used as evaluation metrics. We preprocess the MORPH-II dataset by aligning images first along the average eye position (Raschka, 2018), then by re-aligning so that the tip of the nose is in the center of each image. We do not preprocess the AFAD dataset as faces are already centered. Afterwards, face images are resized to $256 \times 256 \times 3$ and randomly cropped to $224 \times 224 \times 3$ for training. For testing, a center crop of $224 \times 224 \times 3$ is taken.

MORPH-II: This dataset consists of 55,608 face images with age labels between 16 and 70. The dataset is randomly divided into 39,617 training, 4,398 validation, and 11,001 testing images.

AFAD: This dataset consists of 164,432 Asian facial images and age labels between 15 and 40. The dataset is randomly divided into 118,492 training, 13,166 validation, and 32,763 testing images.

**Evaluation metrics:**    MAE (Equation 28) is used as the evaluation metric. We report Cumulative Score (CS$\theta$) for some datasets. CS$\theta$ is defined as the fraction of test images with absolute error less than $\theta$ years.

**Network architecture and training parameters:**    We evaluate our approach on ResNet-50. We perform two runs with different random seeds to determine the learning rate between $[0.00001, 0.0001, 0.001]$ and use a batch size of 64 for all experiments. We use ImageNet pretrained weights to initialize the network. Full training parameters are described in Table 30. We refer to our evaluation on MORPH-II as **AE1** and AFAD as **AE2**.

Table 30: Training parameters for age estimation using MORPH-II and AFAD dataset

| Optimizer | Epochs | Batch size | Learning rate | Learning rate schedule |
|---|---|---|---|---|
| Adam, weight decay=0, momentum=0 | 50 | 64 | 0.0001 | - |

**Related work**    Existing approaches for age estimation include ordinal regression (Niu et al., 2016; Cao et al., 2020), soft regression (Yang et al., 2018), and expected value ordinal regression (Pan et al., 2018; Gao et al., 2018). OR-CNN (Niu et al., 2016) proposed the use of ordinal regression via binary classification to predict the label. CORAL-CNN (Cao et al., 2020) refined this approach by enforcing the ordinality of the model output. SSR-Net (Yang et al., 2018) proposed the use of stage-wise soft regression using the softmax of the classification output to refine the predicted label. MV-Loss (Pan et al., 2018) extended the soft regression approach by penalizing the output of the model based on the variance of the age distribution, while DLDL (Gao et al., 2018) proposed to use the KL-divergence between the softmax output and a generated label distribution to train a model.

We compare BEL with related work in Table 31 and Table 32. 95% confidence intervals are provided.

## D.4 END-TO-END SELF DRIVING

We evaluate our approach on the NVIDIA PilotNet dataset and PilotNet model for end-to-end autonomous driving (Bojarski et al., 2016). In this task, the steering wheel's next angle is predicted from an image of the road. We refer to these experiments as **PN**. MAE (Equation 28) is used as the evaluation metric.

**Dataset**    We use a driving dataset consisting of 45,500 images taken around Rancho Palos Verdes and San Pedro, California (Chen). We crop images to $256 \times 70 \times 3$ then resize them to $200 \times 66 \times 3$. We randomly vary the brightness of the image between $[0.2\times, 1.5\times]$, randomly flip images, and make random minor perturbations on the steering direction. We use $\theta = 10$ with 670 quantization levels for BEL.

Table 31: Age estimation results on MORPH-II dataset (AE1). $\theta = 10$ is used for BEL.

| Approach | Feature extractor | #Parameters (M) | MORPH-II (MAE) | MORPH-II (CS$\theta = 5$) |
|---|---|---|---|---|
| OR-CNN (Niu et al., 2016) (Ordinal regression by binary classification ) | - | 1.0 | 2.58 | 0.71 |
| MV Loss (Pan et al., 2018) (Direct regression) | VGG-16 | 138.4 | 2.41 | 0.889 |
| DLDL-v2 (Gao et al., 2018) (Ordinal regression with multi-class classification) | ThinAgeNet | 3.7 | 1.96* | - |
| CORAL-CNN (Cao et al., 2020) (Ordinal regression by binary classification) | ResNet34 | 21.3 | 2.49 | - |
| Direct Regression (L2 Loss) | ResNet50 | 23.1 | $2.44 \pm 0.01$ | $0.903 \pm 0.002$ |
| BEL-J/BEL-J/BCE | ResNet50 | 23.1 | $\mathbf{2.27} \pm 0.01$ | $\underline{\mathbf{0.928}} \pm 0.001$ |

∗Uses different data augmentation for the training

Table 32: Age estimation results on AFAD dataset (AE2). $\theta = 10$ is used for BEL.

| Approach | Feature extractor | #Parameters (M) | AFAD (MAE) | AFAD (CS$\theta = 5$) |
|---|---|---|---|---|
| OR-CNN (Niu et al., 2016) (Ordinal regression by binary classification ) | - | 1.0 | 3.51 | 0.74 |
| CORAL-CNN (Cao et al., 2020) (Ordinal regression by binary classification) | ResNet34 | 21.3 | 3.47 | - |
| Direct Regression (L2 Loss) | ResNet50 | 23.1 | $3.21 \pm 0.02$ | $0.810 \pm 0.02$ |
| BEL-B1JDJ/GEN-EX/L1 | ResNet50 | 23.1 | $\mathbf{3.11} \pm 0.01$ | $\underline{\mathbf{0.823}} \pm 0.001$ |

**Training parameters** We perform two runs with different random seeds to determine the learning rate between $[0.00001, 0.0001, 0.001]$ and use a batch size of 64 for all experiments. Full training parameters are described in Table 33.

Table 33: Training parameters for end-to-end autonomous driving using PilotNet.

| Optimizer | Epochs | Batch size | Learning rate | Learning rate schedule |
|---|---|---|---|---|
| SGD with weight decay=1e-5, momentum=0 | 50 | 64 | 0.1 | 1/10 at 10, 30 epochs |

**Related work** End-to-end autonomous driving is a novel task that has become increasingly relevant due to the rise of self-driving vehicles. The autonomous driving model's task is to predict the future driving angle based on a forward-facing image from the perspective of the vehicle. PilotNet (Bojarski et al., 2017) used a small, application-specific network to provide good accuracy within the time constraints of autonomous driving.

We compare BEL with the baseline PilotNet architecture in Table 34. 95% confidence intervals are provided.

Table 34: End-to-end autonomous driving results on PilotNet dataset (PN) and architecture (Bojarski et al., 2017; 2016).

| Approach | Feature extractor | #Parameters (M) | MAE |
|---|---|---|---|
| PilotNet (Bojarski et al., 2017) | PilotNet | 1.8 | $4.24 \pm 0.45$ |
| BEL-J/GEN/CE | PilotNet | 1.8 | $\mathbf{3.11} \pm 0.01$ |

