# OpenReview forum: "Label Encoding for Regression Networks"
_ICLR.cc/2022/Conference — ICLR 2022 Spotlight_

### Official Review · Reviewer_HgS5 · 2021-11-01

**Correctness:** 3
**Technical Novelty And Significance:** 3
**Empirical Novelty And Significance:** 3
**Recommendation:** 8
**Confidence:** 3

**Main Review:**

## Pros
- It is interesting to consider such encoding/decoding approaches for regression problems in an application-agnostic setting.
- The end-to-end training using L1/L2 loss with the proposed D^{GEN-EX} seems reasonable, though experimentally "CE + GEN-EX" works better in some cases (D2 vs D4 in Fig.5).
- The error probability of a binary classifier is modeled well with the mixture of Gaussians, and Fig.3(c) gives a nice explanation for the comparison between BEL-U and BEL-J.
- The experiments are conducted with a wide range of regression applications.

## Cons
- I wonder how the best BEL results are selected in Table 2. If they are chosen to minimize the test error, it is not fair to compare them with the results of other approaches. A validation set should be used for model selection.
- For example, HEXJ/GEN-EX/BCE is chosen for PN and its MAE is 3.11 in Table 2, however, the same combination (D5+HEXJ) results in about 5.2 MAE in Fig.5. Why do such discrepancies occur?

## Minor comments
- I would like to see more detailed analysis about the dicision boundaries (red lines in Fig.2) and its relationship to the continuous prediction. It is denoted in p.6 that "Although the hamming distance between 1 and 7 is small, since the differing bits (C1 and C4) are far from decision boundaries and less likely to be mispredicted, a 1 is less likely to be mispredicted as a 7". I can understand the statement, however, it is not clear how such findings affect the prediction.
- In Fig.2, what do NRB mean?
- In Fig.2, I could not understand how B1JDJ and B2JDJ differ in predicting continuous targets.
- In p.7, facial landmark detection seems better abbrebiated by FLD (instead of FDL).
- In p.8, it is denoted that "For HPE4, FDL1, and AE1, Johnson does better than unary." However, I can see no such trend in Fig.5.
- In my opinion, scientific manuscripts should not refer to wikipedia.


**Summary Of The Paper:**

This paper analyzes regression problems in which continuous targets are predicted using a set of binary classifiers. There is a large design space including quantization, encoding, decoding, and loss functions. An expected value-based function (D^{GEN-EX}) is proposed as a new decoding function.

The authors apply various combinations of these functions and compare them theoretically and experimentally. It is suggested that no single combination outperforms the others, but the proposed BEL approach has a potential to improve the prediction accuracy of continuous regression problems.


**Summary Of The Review:**

Overall, I vote for weak rejection. I like the motivation of the work and the authors provide a nice analysis of the problem from both theoretical and empirical perspectives. My major concern is about the clarity of the experiments in which the proposed BEL approach is compared with the existing regression models, as mentioned in the main review above.

=====POST-REBUTTAL COMMENTS========

Thanks for the authors' response. The newly added experimental results and manuscript modifications address all of my concerns. I have raised my score to recommend this paper to be accepted.

---

> ### Author Response · Authors · 2021-11-17
> **Author response (part 2)**
>
>
> **2.  For example, HEXJ/GEN-EX/BCE is chosen for PN and its MAE is 3.11 in Table 2, however, the same combination (D5+HEXJ) results in about 5.2 MAE in Fig.5. Why do such discrepancies occur?**
>
> ***Changes in the paper:*** That is a typo in Table 2, and the best combination for PN should be J/GEN/CE. We have corrected it in Table 2.
>
> **3.  Detailed analysis about the decision boundaries and its relationship to the continuous prediction.**
>
> Although any classification error results in an erroneous binary code and thus an erroneous decoded continuous prediction, the probability of misclassification is higher for bits closer to decision boundaries (Figure 3a and 3b). Ideally, for those bits that are likely to be mispredicted, the erroneous continuous prediction should change little from the target (i.e., error-free decoded value). As we want to minimize the expected change in decoded prediction given the probability of classification errors, we cover the impact of decision boundaries on the continuous predictions in the second design property proposed in Section 3.2.
>
> According to the second design property, a suitable code design should maximize the distance between two faraway values such that the differing bits between two codes are distant from decision boundaries. Meeting this property ensures that the decoded continuous prediction is not far away from the target value. An example of this is given for the Johnson code in Section 3.2. Here all differing bits between 1 and 7 ($b^1$ and $b^4$) are far from decision boundaries and less likely to be mispredicted. This means that for a target value 1, the probability of erroneous output code being decoded to a continuous value 7 is very low, thus reducing the expected error between target and predicted continuous values.
>
> ***Changes in the paper:*** We have modified the description of the Johnson code.
>
> **4. In Fig.2, what do NRB mean?**
>
> NRB is the abbreviation of non-reflected binary. For B1JDJ and B2JDJ codes, we adapt reflected binary codes to minimize the misclassification probability around the design boundary for the radix-term. Figure 2d gives an example of B1JDJ code without reflected binary in radix term $(b^2 b^3)$. We provide the results for B1JDJ without reflected binary in Appendix A to demonstrate the effectiveness of using reflected binary for the code design.
>
> ***Changes in the paper:*** We have clarified the meaning of NRB in Figure 2 caption.
>
> **5.  In Fig.2, I could not understand how B1JDJ and B2JDJ differ in predicting continuous targets.**
>
> B1JDJ uses one bit for the base term and thus needs (N/4 +1) bits to represent N quantization levels. Whereas B2JDJ uses two bits for base and needs (N/8 + 2) bits to represent N quantization levels. In the previous draft, for the given example with N=8 in Figure 2, both B1JDJ and B2JDJ result in the same code length and codes. We have modified Figure 2 to give an example for the range $1-16$ that clarifies the difference between B1JDJ and B2JDJ codes.
>
> ***Changes in the paper:*** Figures 2e and 2f are modified with new examples of B1JDJ and B2JDJ codes.
>
> **6.  In p.8, it is denoted that "For HPE4, FDL1, and AE1, Johnson does better than unary." However, I can see no such trend in Fig.5.**
>
> ***Changes in the paper:*** We have clarified that for D1 (decoding functions same as used in Section 3.1 for comparison of Johnson and unary code), this trend is observed for HPE3, FLD1, and AE1 in Section 4.
>
> **7.  In my opinion, scientific manuscripts should not refer to Wikipedia.**
>
> ***Changes in the paper:*** Thank you for the suggestion. We have edited references.

---

> > ### Comment · Reviewer_HgS5 · 2021-11-18
> > **Thanks for the response**
> >
> > Thanks for your thorough response. The updated experimental results strongly suggest the potential of BEL. I have modified my review and raised score.

---

> ### Author Response · Authors · 2021-11-17
> **Author response (part 1)**
>
> Thank you for your thoughtful comments and feedback. We have updated the paper and supplemental material to reflect the response below.
>
> **1.  The selection method for the best BEL results:**
>
> The datasets used for the benchmarks evaluated in the paper do not provide separate validation sets; thus, the test set was used for model selection for results reported for all regression approaches in Table 2, including the earlier works on specialized approaches.
>
> We agree that a validation set should be used for model selection. Hence we have reevaluated the benchmarks with 20% of the training data being used as a validation set to select the best model. We have reported updated results in which the validation error is used to choose the best BEL encoding/decoding/loss function combination. We report results for two setups.
>
> 1.  In the first setup, we use 80% of the training data for training and use validation error to select the best BEL approach and for model selection. Since earlier works use 100% training data for the reported results and use test error for model selection (i.e., which model is the best over multiple epochs), we have re-run specialized approaches (if possible), direct regression, and multiclass classification. It was not possible for us to re-run experiments for all specialized approaches with $80\%$ dataset due to resource constraints, and the comparison is conservative for many benchmarks since we use less data while training for the BEL approach than prior works used for model selection.
>
> 2.  Earlier works on specialized approaches use the complete training dataset for training. For a better comparison, in the second setup, we select the BEL approach to use based on validation error and train our model on 100% of the training data.
>
> The table below provides errors (MAE or NME) for different benchmarks based on this setup.
> Note that the additional results do not diminish the effectiveness of BEL and BEL outperforms direct regression and multiclass classification for all benchmarks and specialized approaches for several benchmarks. The optimal BEL combination selected based on validation remains the same for several benchmarks.
>
> |  | HPE1 | HPE2 | HPE3 | HPE4 | FLD1 | FLD2 | FLD3 | FLD4 | AE1 | AE2 | PN |
> |---|:---|:---|:---|:---|:---|:---|:---|:---|:---|:---|:---|
> | BEL E/D/L functions | U/GEN-EX/L2 | U/GEN-EX/BCE | B1JDJ/GEN-EX/BCE | U/GEN-EX/BCE | HEXJ/GEN-EX/CE | U/GEN-EX/CE | B1JDJ/GEN-EX/CE | B1JDJ/GEN-EX/CE | J/BEL-J/BCE | B1JDJ/GEN-EX/L1 | J/GEN/CE |
> | Specialized approach (Re-run on 80% training set) | - | - | 4.08+-0.11 | 4.69+-0.02 | - | - | - | - | - | - | - |
> | Direct regression (80% training set) | 6.12+-0.02 | 5.97+-0.09 | 4.08+-0.11 | 4.69+-0.02 | 3.70+-0.04 | 3.69+-0.06 | 4.71+-0.02 | 1.51+-0.01 | 2.45| 3.34 | 4.56 |
> | Multiclass classification (80% training set) | 5.38+-0.03 | 5.60+-0.13 | 5.58+-0.04 | 5.86+-0.10 | 3.64+-0.02 | 3.68+-0.02 | 4.77+-0.02 | 1.56+-0.01 | 2.85 | 3.47 | 6.37 |
> | BEL (80% training set) | 3.91+-0.08 | 4.91+-0.10 | 3.50+-0.08 | 3.99+-0.04 | 3.35+-0.02 | 3.40+-0.03 | 4.37+-0.01 | 1.48+-0.01 | 2.36 | 3.20 | 3.49 |
> | Specialized approach (Result repored in exising works and trained on full training set) | - | - | 3.40 | 4.14 | 3.45 | 3.32 | 4.32 | 1.57 | 2.49 | 3.47 | 4.24 |
> | Direct regression (Full training set) | 4.76+-0.35 | 6.71+-0.13 | 3.40+-0.26 | 4.14+-0.12 | 3.60+-0.02 | 3.54+-0.03 | 4.64+-0.03 | 1.51+-0.01 | 2.44+-0.01 | 3.21+-0.02 | 4.24+-0.45 |
> | Multiclass classification (Full training set) | 4.49+-0.24 | 5.31+-0.05 | 5.22+-0.04 | 5.14+-0.08 | 3.58+-0.03 | 3.36+-0.02 | 4.50+-0.01 | 1.56+-0.01 | 2.76+-0.03 | 3.38+-0.05 | 5.54 |
> | BEL (Full training set) | 3.56+-0.01 | 4.77+-0.05 | 3.30+-0.04 | 3.9+-0.03 | 3.34+-0.02 | 3.40+-0.02 | 4.36+-0.02 | 1.47+-0.00 | 2.27+-0.01 | 3.11+-0.00 | 3.11+-0.01 |
>
> ***Changes in the paper:*** We have modified Section 4 and Table 2 to include results based on validation error used to choose the best BEL approach. We have also added evaluation on $80\%$ training data in Appendix A.

---

### Official Review · Reviewer_qL8F · 2021-11-02

**Correctness:** 3
**Technical Novelty And Significance:** 3
**Empirical Novelty And Significance:** 3
**Recommendation:** 6
**Confidence:** 3

**Main Review:**

- as far as i understood the regression network and the decoding function are trained end-to-end. It is unclear how to perform end-to-end training. what is the total loss?
- is the proposed method applicable to the prediction models other than regression networks? (although end-to-end training may not be available)
- it would be interesting to see the effect of skewness, multimodality, and outliers of the real-valued label on the effectiveness of the proposed method. they needs to be investigated to demonstrate the robustness of the method.
- also, how does the scale of the original label and any normalization of the label on the proposed method?
- for each dataset how did you choose the hyperparameters for label encoding? e.g. which encoding/decoding functions to choos, quantization level and theta. How they can be determined for a new dataset? Although it seems they were empirically determined in the experiments, it would be great if a guideline is suggested.
- for readers' convinence, please describe what each notation is in the main body of the manuscript. (e.g. theta in Table 1 is only described in appendix)

**Summary Of The Paper:**

This study presents a label encoding method (binary encoded labels) to transform a regression problem into multiple binary classifiation problems. The authors designed encoding and decoding functions to encode a real-value into multiple classes and decode the classes into the real value. The proposed demonstrated improved performance on benchmark datasets.


**Summary Of The Review:**

The idea is interesting. The method is simple yet effective. I think the some issues needs to be clarified.

---

> ### Author Response · Authors · 2021-11-17
> **Author response (part 2)**
>
> **4.  Effect of scale and normalization of the label on the proposed method:**
>
> As the scale of the original label increases, the number of quantization levels needs to be increased for BEL to reduce the quantization error. We propose more compact codes such as HEXJ that can be used for a large number of quantization levels without increasing the number of binary classifiers linearly.
>
> *Effect of normalization:*
>
> In BEL, the network is trained to predict a binary code corresponding to the encoded labels. Real values of the labels are not used in BEL when BCE or CE loss functions are used, but only the quantization levels are used. In this case, any linear normalization does not affect BEL as long as the number of quantization levels is kept fixed.
> The real-valued labels are only used for training BEL when L1/L2 loss function is used, and in this case, a normalization technique to reduce the range of labels (e.g., scaling label y to y/t) only affects the magnitude of the loss and but not the target values of output logits. As shown in the table, the scaling factor has a negligible impact on errors.
>
> | Scaling factor | t=1  | t=10  | t=256 |
> |---|:---:|:---:|:---:|
> | Error (NME) for FLD1 (U/GEN-EX/L1) | 3.37 | 3.38 | 3.38 |
>
> **5.  Selection of hyperparameters for label encoding:**
>
> This work mainly focuses on studying the impact of encoding, decoding, and loss functions for binary-encoded labels. In our experiments, we have set $\theta$ to $10$ or $30$ depending on the model size increase. We observe empirically that for $150-300$ quantization levels, $\theta=10$ or $\theta=30$ can be used for simpler (e.g., U and J) or complex (e.g., B1JDJ/B2JDJ/HEXJ) encoding functions, respectively. We have chosen quantization levels based on the numeric range of real-valued labels and kept the quantization step equal to one for all benchmarks. We have performed a limited exploration of $\theta$ and quantization and mainly focus on encoding, decoding, and loss functions.
>
> We evaluate different BEL encodings using a validation set and select the encoding/decoding/loss functions based on this. As shown in the evaluation, the exploration of limited encoding, decoding, and loss functions shows significant improvement in accuracy compared to direct regression. We observe that for problems with high numbers of labels (e.g., WFLW), more compact codes such as B1JDJ/B2JDJ/HEXJ work better. We agree that finding what combination is suitable for a new task is an interesting problem. However, Our submission aimed to find to what extent the choice of encoding/decoding functions affects the accuracy of regression tasks.

---

> ### Author Response · Authors · 2021-11-17
> **Author response (part 1)**
>
> Thank you for your thoughtful comments and feedback. We have updated the paper and supplemental material to reflect the response below.
>
> **1.  As far as I understood the regression network and the decoding function are trained end-to-end. It is unclear how to perform end-to-end training. what is the total loss?**
>
> In our experimental setup, the feature extractor and regressor are trained end-to-end. The decoding functions are kept fixed and are not trained for BEL.
>
> We explore three types of decoding functions in this work. Out of these, $\mathcal{D}^{\text{GEN}}$ and $\mathcal{D}^{\text{GEN-EX}}$ decoding functions can be used for the loss calculation and minimize the mismatch between decoded predictions and target values directly.
>
> 1.  $D^{GEN}$: Decoding function ${D}^{\text{GEN}}$ finds the correlation between each row  $C_{k, :}$  of the code matrix $C$ and the output $\hat{Z}_i$. $C \hat{Z}_i$ gives the correlation vector, and the index with the highest correlation is used as the predicted label.
>
>       In this case, cross-entropy loss ${L}_{\text{CE}} \big(C{ } \hat{Z}_i, Q_i\big)$ can be used to train the network.
>
>
> 2.  $D^{GEN-EX}$: This decoding function is differentiable and gives a real-valued decoded prediction. Thus L1/L2 loss function $L(D^{GEN-EX}(Z_i),y_i)$ can be used to train the network.
>
> ***Changes in the paper:*** We have clarified the use of different loss and decoding functions in Section 3.3.
>
> **2.  is the proposed method applicable to the prediction models other than regression networks? (although end-to-end training may not be available)**
>
> We assume that the question is about using the proposed method for prediction models other than deep learning-based models. Please let us know if we misunderstood your question.
>
> In this work, we focus on studying different encoding and decoding functions for deep learning-based regression models. However, our insights on desirable properties for suitable encoding functions and decoding functions are applicable regardless of the underlying prediction model. We believe that the proposed generalized framework based on encoding and decoding functions to reduce regression problems to a set of binary classification problems and propose encoding and decoding functions can be used for the other prediction models.
>
> **3.  Effect of skewness, multimodality, and outliers of the real-valued labels:**
>
> We have added the plots for the distribution of real-valued labels for several benchmarks in [this link](https://drive.google.com/file/d/12svka0R0lV873fKIhrPNlRyXXfu-ACBH/view?usp=sharing).
>
> Although we have not directly investigated the impact of skewness, multimodality, or outliers on the effectiveness of the proposed method, some of the datasets we study exhibit these properties. We evaluate the proposed approach on 11 benchmarks, where several benchmarks have more than one label (e.g., FLD3 has 98 labels). We observe that our benchmarks cover different distributions of labels, where several benchmarks exhibit skewness, multimodality, or outliers in the distribution of labels. For example, the AFAD dataset exhibits high skew, the MORPH-II dataset exhibits multimodality, and the PilotNet dataset has a very long tail distribution with many outliers. Our evaluation suggests that BEL is robust to these characteristics of label distribution and achieves significant accuracy improvement over direct regression for these benchmarks.
>
> The proposed encoding and decoding functions for BEL are based on the error analysis and error probability distribution of the binary classifiers in Section 3.1. We observe that even for datasets with skewed label distribution, the error probability distribution of binary classifiers is close to the proposed Gaussian distribution. Thus the proposed encoding and decoding functions apply to such datasets too.

---

### Official Review · Reviewer_zpsJ · 2021-11-03

**Correctness:** 3
**Technical Novelty And Significance:** 3
**Empirical Novelty And Significance:** 3
**Recommendation:** 8
**Confidence:** 4

**Details Of Ethics Concerns:**

I have none.

**Main Review:**

# A. Strengths

1. **Novelty.**
  Error-correcting codes are used in ML for classification tasks since at least 1995. To the best of *my* knowledge, the authors are the first to propose using such codes in regression tasks. This is interesting.

1. **Reported empirical performance seems much better than other methods.**
The results speak for themselves (especially in Table 2).
I should say that I am not an expert in deep regression tasks, and cannot say whether the methods the authors compare to (e.g. Ruiz et al. 2018 or Behera et al. 2021) are sufficient.
Moreover, I have some reservations regarding the comparison to the simplest baseline (the direct regression baseline) which I address below.

# B. Weaknesses

1. **Empirical advantage over other methods should be more carefully demonstrated.**
Since the reported results are *remarkable* and are claimed to be SOTA, I feel like "extraordinary claims require extraordinary evidence".
Importantly, no code is provided, thus I feel even more obligated to verify that the comparisons are fair.
Especially, I am bothered by the comparison to the "direct regression" baseline.
It seems like the feature extractors' parameters were *frozen* when comparing direct regression to the proposed BEL methods.
In this case, I suspect the added linear layer (reduction to $\theta$ features) significantly helps the model, and that its absence from the direction regression models makes the comparison *unfair*.
I would even say that I would be slightly surprised if when the pretrained feature extractors will be also learned (rather than frozen), the direction regression approach will not be competitive with the proposed BEL method.
I therefore decrease my general score, until more evidence and details are provided.

2. **Readability issues.**
Throughout my reading, several points were not very clear, and it made the reading slower and more difficult than actually needed.
For instance,
* The experimental part is especially hard to understand. It was unclear what method exactly is the "Multiclass classification" one. Appendices D-G are also hard to interpret. The reader is expected to cross between many tables that use different names for the same methods. For instance, see Table 11. How can one distinguish between results for HPE2 and results for HPE4?
* Usage of the "decision boundaries" term is untraditional and can be confusing.
* Description of the HEXJ encoding is hard to understand.
* Paper says "We evaluate the Hadamard code to show that..." but the results are not actually shown anywhere.
* Figure 1b is hard to understand, especially on B&W hard copies.


3. **Analytic results do not contribute much.**
Section 3.1 offers some basic analysis of two encoding-decoding combinations (using the Unary and Johnson codebooks).
I am not sure what is the contribution of this analysis to the rest of the paper or to the readers' understanding.

# C. Other comments / thoughts

1. I allow myself to assume the Hadamard codebook fails because each column has many "bit flips" (i.e. decision boundaries in the authors' terminology). This implies that traditional classification codebooks (e.g. Hadamard and random ones) are not suitable for regression tasks.

1. Using a fine quantization, codebooks like the unary or Johnson codebooks will largely "inflate" the output space. The authors' HEXJ codebook hints that the output space does not have to grow *linearly* with the number of quantization "buckets". This is related to the use of ECOC for extreme classification (see *Learning compact class codes for fast inference in large multi class classification* [Cisse et al. 2012] and *Efficient Loss-Based Decoding on Graphs for Extreme Classification* [Evron et al. 2018]). A brief discussion in that direction can be interesting.

1. In the second paragraph of Section 3.2, the statement "we note that a Hamming distance does not..." should be justified.

1. Just before equation (1), should it be "for $k>Q_i$ instead of $k<Q_i$"?

1. The decoding function in equation (1) is not really intuitive and it feels unmotivated given the structure of the unary code.

1. In Table 2, the best FDL2 method is the specialized one, not BEL.

**Summary Of The Paper:**

The paper tackles regression problems using an error-correcting code (ECOC) approach, i.e., reducing a *regression* problem into multiple binary *classification* problems.
ECOC have been so far mainly used for classification tasks, and their application to regression problems in this paper is elegant and seems novel.
The paper examines several combinations of (encoding method $\times$ decoding method $\times$ loss function) on several (deep) regression tasks and reports significant improvements over existing methods.
Some brief analytic discussion is made.

**Summary Of The Review:**

Idea is novel and interesting.
Results and findings are mainly empirical and look remarkable, achieving SOTA performance.
I believe these findings require more careful description of the exact empirical process, and that as a reviewer I should make sure the comparison to other baselines is fair. A provided code would also be helpful and more assuring.

---

> ### Author Response · Authors · 2021-11-17
> **Author response (part 3)**
>
> **13.  In Table 2, the best FDL2 method is the specialized one, not BEL.**
>
> We are sorry for the typo in Table 2. We have corrected Table 2.
>
> **References:**
>
> [1] Ley Xie, Huifang Chen, Peiliang Qiu, and Ming Zhang, "The modified Hamming bound for unequal error protection codes," IEEE 2002 International Conference on Communications, Circuits, and Systems and West Sino Expositions, 2002, pp. 92-95 vol.1, DOI: 10.1109/ICCCAS.2002.1180579.
>
> [2] Wu, Chai Wah. “Designing communication systems via iterative improvement: error correction coding with Bayes decoder and codebook optimized for source symbol error.” ArXiv abs/1805.07429 (2018).

---

> > ### Comment · Reviewer_zpsJ · 2021-11-21
> > **Raising my score**
> >
> > After reading the other reviews and the authors' detailed responses, I decided to increase my score for the latest version of the paper.
> > Good luck.

---

> ### Author Response · Authors · 2021-11-17
> **Author response (part 2)**
>
> **5.  Results for the Hadamard code in the paper**
>
> We did not include errors for the Hadamard code in the evaluation (Figure 5) as the error (MAE or NME) for the Hadamard code is significantly higher than other codes. Further, the Hadamard code does not fall within any of our code design parameters. Due to space constraints, we include results for the Hadamard code in Appendix A (Table 3-13).
>
> ***Changes in the paper:*** We have clarified this is the paper and added a reference to Appendix A,  where results for Hadamard codes are provided.
>
> **6.  Figure 1b is hard to understand, especially on B&W hard copies.**
>
> ***Changes in the paper:*** We have removed this figure from the draft.
>
> **7.  Contribution of the analytic results**
>
> The analytical error analysis in Section 3.1 mainly focuses on understanding the relationship between regression error and classification error for ordinal labels using simple encoding and decoding functions. The observations from the analytical study and error probability distribution of binary classifiers are used to derive the desirable properties of encoding and decoding functions explored in this work. Furthermore, analytical comparison of unary and Johnson code also provides insights for empirically observed trends, as unary code outperforms Johnson code in some benchmarks and vice versa.
>
>
> ***Changes in the paper:*** We have modified Section 3.1 to clarify the motivation behind the analytical study.
>
> **8.  Traditional classification codebooks (e.g. Hadamard and random ones) are not suitable for regression tasks.**
>
> Yes, that is correct. We find that traditional codebooks used for classification such as Hadamard and random codes are not suitable for regression tasks for two reasons:
>
> 1.  Each column has many bit transitions, resulting in a complex function to be learned by each binary classifier.
>
> 2.  Such codes do not consider the objective and properties of the regression tasks. Thus we propose desired properties for suitable code based on our analytical study and observation for regression. Hadamard code does not follow these design properties and results in higher errors for regression tasks.
>
> ***Changes in the paper:*** We have modified the description of the Hadamard code to explain the reason for its unsuitability for regression tasks in Section 3.2.
>
> **9.  Related work on extreme classification:**
>
> Thank you for pointing out these works. Similar to these works, the intuition behind the HEXJ code is to eliminate a linear increase in the number of binary classifiers with more quantization levels. HEXJ is designed to achieve this while being suitable for regression tasks. We have added a brief discussion on this in the related work section.
>
> ***Changes in the paper:*** A brief discussion is added in Section 2 (related work).
>
> **10.  In the second paragraph of Section 3.2, the statement "we note that a Hamming distance does not..." should be justified.**
>
> Hamming distance measures the number of bits changes between two binary strings. Since hamming distance weighs all the bits equally, hamming distance-based codes provide error protection capability to all the bits equally (Wu, 2018; Xie et al., 2002) and do not account for which bits are more likely to be erroneous.
>
> ***Changes in the paper:*** We have expanded the description and added references in Section 3.2
>
> **11.  Just before equation (1), should it be "for $k>Q_i$ instead of $k<Q_i$"?**
>
> We believe that it should be $k<Q_i$. For example, in Figure 2b, for $Q_i=5$, the binary code ($b^1,b^2,b^3, ...,b^7$) is (1111000). Here $b^k$ for $k<5$ is equal to $1$, else $0$. The earlier version of Figure 2 did not clearly show $(b^1,b^2,b^3,...,b^7)$. We have updated the figure.
>
> ***Changes in the paper:*** We have modified Figure 2 to indicate $b^k$ to clarify the relation between $Q$ and $B$.
>
> **12.  The decoding function in equation (1) is not really intuitive and it feels unmotivated given the structure of the unary code.**
>
> Given the structure of the Unary code, different decoding functions can be defined. For example, a reasonable decoding rule can be $\mathcal{D}(\hat{B}_i) = max${$k:\hat{b}^k_i=1$}$ + 1$. However, such a decoding rule can result in a large regression error compared to the one proposed in equation (1). For example, for a target code $1100000$ (i.e., target $Q_i=3$), if the output of classifiers is $\hat{B_i}=1100010$ with an error in $b^6$, the above decoding function would result in decoded value equal to $7$. In contrast, the decoding function in equation (1) would decode it to $4$, which is closer to the target value $Q_i=3$. For the decoding function in equation (1), a single-bit error in the predicted code can change the decoded predicted value by only one quantization level. Thus we use the decoding function defined in equation (1).
>
> ***Changes in the paper:*** The motivation behind the use of equation (1) is added in Section 3.1.

---

> ### Author Response · Authors · 2021-11-17
> **Author response (part 1)**
>
> Thank you for your thoughtful comments and feedback. We have updated the paper and supplemental material to reflect the response below.
>
>  **1. The empirical advantage over other methods should be more carefully demonstrated.**
>
> We had to wait until the discussion forum was opened to post a reviewer-visible comment and keep the code visible only to the reviewers and ACs. We apologize for the confusion. We have provided the link for the code implementation for direct regression, multiclass classification, and binary encoded labels in a reviewer- and AC-visible comment and have provided training and inference code with trained models. We will make the code publicly available if accepted.
>
> We would like to clarify that the feature extractors’ parameters are not frozen when training for direct regression. The entire network (feature extractor regressor) is trained end-to-end for direct regression, multiclass classification, and binary encoded labels. We have clarified this in Section 4 in the updated draft.
>
> Furthermore, we have also conducted experiments for direct regression and multiclass classification with two fully connected layers (added linear layer with $\theta$ similar to BEL). We observe that increasing the number of fully connected layers does not significantly impact the accuracy for direct regression or multiclass classification (except AE1 and AE2), and in fact, degrades the accuracy for most benchmarks (possibly due to overparmeterization). Our evaluation shows that BEL outperforms direct regression and multiclass classification even when normalizing for layer count. We would like to point out that due to the increased number of output logits with BEL, the number of parameters used for a model with BEL will generally be slightly greater than one trained using a single output logit with direct regression. However, since the model size is dominated by the feature extractor, this increase is negligible (see Table 2 to compare model sizes of different approaches).
>
>   | Benchmark | Direct regression |Direct regression  | Multiclass classification | Multiclass classification | BEL |
> |:---:|:---:|:---:|:---:|:---:|---|
> | | 1 FC layer | 2 FC layers | 1 FC layer | 2 FC layers | 2 FC layers |
> | HPE1 | 4.76 | 5.19 | 4.49 | 4.82 | 3.37 |
> | HPE2 | 5.65 | 5.59 | 5.31 | 5.42 | 4.77 |
> | HPE3 | 3.40 | 3.54 | 4.45 | 4.54 | 3.12 |
> | HPE4 | 4.14 | 4.22 | 5.14 | 5.45 | 3.90 |
> | FLD1 | 3.6 | 3.63 | 3.58 | 3.56 | 3.34 |
> | FLD2 | 3.54 | 3.58 | 3.51 | 3.62 | 3.36 |
> | FLD3 | 4.64 | 4.63 | 4.50 | 4.64 | 4.33 |
> | FLD4 | 1.51 | 1.51 | 1.56 | 1.53 | 1.47 |
> | AE1 | 2.44 | 2.35 | 2.75 | 2.81 | 2.27 |
> | AE2 | 3.21 | 3.14 | 3.38 | 3.40 | 3.11 |
> | PN | 4.24 | 4.33 | 4.56 | 5.74 | 3.11 |
>
> ***Changes in the paper:*** Clarification about the training method for direct regression is added in Section 4.  An ablation study on the effect of more fully connected layers on accuracy is added in Appendix A (Page 19).
>
> **2. Multiclass classification for regression**
>
> In the multiclass classification-based regression, the target values are quantized and converted to a class. The network is trained using cross-entropy loss in this case.
>
> ***Changes in the paper:*** We have added a description in Section 4.
>
> **3. Usage of the "decision boundaries" term is untraditional and can be confusing.**
>
> We agree that this term is used in an untraditional way in the draft and can be confusing. We used the term decision boundary to signify the transitions $0->1$ and $1->0$ in the binary classifier’s target output over the numeric range of labels. We have modified the draft to define and use the term “bit transition” to represent these transitions. For a binary classifier, a decision boundary separates the examples from two different classes.  In the context of using binary classifiers for regression using BEL, a higher number of bit transitions results in a more complex function for the decision boundary to be learned by the binary classifier.
>
> ***Changes in the paper:*** We have replaced the term “decision boundary” with “bit transition” to represent the transitions $0->1$ and $1->0$ in the binary classifier’s target output over the numeric range of labels.
>
> **4. Description of the HEXJ encoding is hard to understand.**
>
> A number can be encoded by converting each digit to a fixed binary code. Following this approach, we propose HEXJ, in which each digit ($0-F$) of the hexadecimal representation of a number is converted to an 8-bit binary code using Johnson code. For example, for the decimal number $47$ (i.e., $2F$ in hex), HEXJ($47$) = Concetanate(Johnson($2$), Johnson($F$)). A 16-bit HEXJ code can represent numbers in the range of $00$ to $FF$ (a total of $256$), reducing the number of classifiers significantly for a large number of quantization levels.
>
> ***Changes in the paper:*** We have modified the description of the HEXJ code in Section 3.2.

---

### Official Review · Reviewer_t2zZ · 2021-11-04

**Correctness:** 3
**Technical Novelty And Significance:** 3
**Empirical Novelty And Significance:** 3
**Recommendation:** 8
**Confidence:** 3

**Main Review:**

Strengths:
- The paper is detailed with a lot of information useful for the reproduction of the experiments.
- The theory of the binary-encoded label is well described also with clarifying examples.
- The analysis of Encoding/Decoding functions has been conducted for a simple Unary coding (BEL-U) and the Johnson coding (BEL-U) (in the Appendix).
- The authors introduced an estimation of the error probability for BEL-J and BEL-U and an interesting relation between the two methods (in terms of the percentage difference between errors).
- Even though the usage of classification for regression problem is not original, the work try to shed lights on some design choices.

Weaknesses:
- The paper is generally well written but I suggest to revise the organization because some parts are introduced after they are effectively used and this can make confusion.
For example:
  - The Unary code, Johnson Code and other Encodings are described on 3.2 but Unary and Johnson are used already in 3.1. I suggest to anticipate the description of the encoding functions or at least for Unary and Johnson ones.
  - The name of BEL-J decoding is not introduced neither in 3.1 nor in 3.3 but in the appendix and it is used in the caption of Figure 5.
- For the two detailed codes (Unary and Johnson), I suggest to describe clearly the Encoding and Decoding functions and then the computation of the error and other aspects present in sections 3.1, 3.2 and 3.3. For this reason I think that the Sections 3.1, 3.2, 3.2 should be revised in order to improve readability.
- According to Figure 4, the authors seems to have used only 1 Fully connected layer to connect feature vector to the output. In the Figure 13, there is a "feature extraction extension" but, generally speaking, it seems that only a shallow network is present between the feature extractor and the output layer. In this situation the using of binary-encoded classifier seems to present some advantages.
Unfortunately, no tests have been performed to understand if increasing the number of Fully connected layers impact on the results.
- The BEL approaches outperform the other tested approaches but the best design parameters depend on task and dataset. As authors said this is left for the future works but the doubt is that with more layers the situation could change.


Other comments:
- Figure 1(a) the caption should be "(a) Training (top) and inference (bottom) flow"
- Instead of use "MAE/NME" I suggest to use "Error (MAE or NME)" as done in Table 2 since the currently used name could be misleading.
- Why in Table 2 some approaches (all specialized approach and some direct regression) don't have the indication of standard deviation?
- Caption of Figure 5 should be improved.

Some typos:
resepct --> respect
deocding --> decoding

**Summary Of The Paper:**

The paper describes a general framework to solve a regression problem as a binary classification.
The labels (targets) are binary encoded and hence the regression problem is solved using a set of binary classifiers.
Since the process is dependent by the type of Encoding and Decoding used, the authors compare different type of configurations showing, empirically, the effect of different design parameters on the final accuracy.

**Summary Of The Review:**

The authors propose a general framework to approach regression problems using binary encoding and a set of classifiers. The paper is detailed, some interesting theoretical aspects are discussed and experiments seem to demonstrate the high accuracy of proposed approach even though only using shallow network after the feature extractor.

=====POST-REBUTTAL COMMENTS========

The added experimental results and manuscript modifications address all of my concerns.
I have raised my score.

---

> ### Author Response · Authors · 2021-11-17
> **Author response**
>
> Thank you for your thoughtful comments and feedback. We have updated the paper and supplemental material to reflect the response below.
>
> **1. Impact of increasing the number of fully connected layers on the results:**
>
> We perform an ablation study to evaluate the effect of the number of fully connected layers on the error for direct regression and multiclass classification. The table below provides the error (MAE or NME) for direct regression and multiclass classification with one or two fully connected layers after the feature extractor. We observe that increasing the number of fully connected layers in direct regression and multiclass classification does not improve the accuracy for most benchmarks (possibly due to overparameterization). BEL with two fully connected layers outperforms direct regression and multiclass classification in both cases; thus increasing the number of fully connected layers does not impact the suitability of binary encoded labels.
>
> | Benchmark | Direct regression | Direct regression| Multiclass classification | Multiclass classification  | BEL |
> |:---:|:---:|:---:|:---:|:---:|---|
> | | 1 FC layer | 2 FC layers | 1 FC layer | 2 FC layers | 2 FC layers |
> | HPE1 | 4.76 | 5.19 | 4.49 | 4.82 | 3.37 |
> | HPE2 | 5.65 | 5.59 | 5.31 | 5.42 | 4.77 |
> | HPE3 | 3.4 | 3.54 | 4.45 | 4.54 | 3.12 |
> | HPE4 | 4.14 | 4.22 | 5.14 | 5.45 | 3.90 |
> | FLD1 | 3.6 | 3.63 | 3.58 | 3.56 | 3.34 |
> | FLD2 | 3.54 | 3.58 | 3.51 | 3.62 | 3.36 |
> | FLD3 | 4.64 | 4.63 | 4.50 | 4.64 | 4.33 |
> | FLD4 | 1.51 | 1.51 | 1.56 | 1.53 | 1.47 |
> | AE1 | 2.44 | 2.35 | 2.75 | 2.81 | 2.27 |
> | AE2 | 3.21 | 3.14 | 3.38 | 3.40 | 3.11 |
> | PN | 4.24 | 4.33 | 4.56 | 5.74 | 3.11 |
>
> ***Changes in the paper:*** Ablation study on the effect of the number of fully connected layers on error is added in Appendix A (Page 19).
>
> **2. Impact of more layers on the binary-encoded labels:**
>
> We believe the question is regarding the effect of more fully connected layers on BEL; please let us know if we misunderstood. The table below shows the impact of increasing the number of fully connected layers on the error (MAE or NME) for different regression approaches. We observe that increasing the number of fully connected layers degrades the accuracy in most cases. Furthermore, even for more fully connected layers in BEL, different encoding functions result in the lowest error.
>
> | Benchmark | # FC layers (size of FC layers) | Direct regression | Multiclass classification | U/BCE/GEN-EX | J/BCE/GEN-EX | B1JDJ/BCE/GEN-EX | B2JDJ/BCE/GEN-EX | HEXJ/BCE/GEN-EX |
> |---|---|---|---|---|---|---|---|---|
> | FLD1 | 1 (1024-x) | 3.6 | 3.58 | - | - | - | - | - |
> | | 2 (1024-30-x) | 3.63 | 3.56 | 3.45 | 3.43 | 3.42 | 3.41 | 3.47 |
> | | 3 (1024-30-10-x) | 3.63 | 3.94 | 3.55 | 3.47 | 3.82 | 4.02 | 3.62 |
> | FLD2 | 1 (1024-x) | 3.54 | 3.51 | - | - | - | - | - |
> | | 2 (1024-10-x) | 3.58 | 3.62 | 3.48 | 3.46 | 3.43 | 3.42 | 3.38 |
> | | 3 (1024-30-10-x) | 3.55 | 3.78 | 3.42 | 3.46 | 3.5 | 3.61 | 3.52 |
>
> ***Changes in the paper:*** Ablation study on the effect of the number of fully connected layers on error is added in Appendix A (Page 19).
>
> **3. The standard deviation for some approaches in Table 2:**
>
> The direct regression results for HPE3 and HPE4 are reported from the earlier work by Behera et al., 2021, which does not include standard deviation in the reported results. We have reproduced their results and added standard deviation in the updated draft.
>
> ***Changes in the paper:*** We have added standard deviation for direct regression (HPE3 and HPE4) in Table 2.
>
> **4. Organization of Section 3:**
>
> ***Changes in the paper:*** We have modified Section 3 to include the suggestions.

---

> > ### Comment · Reviewer_t2zZ · 2021-11-19
> > **Raised Score**
> >
> > Thank you for your response.
> > The updated experimental results address my concerns.
> > I raised score consequently.

---

### Decision · Program_Chairs · 2022-01-20

**Decision:**

Accept (Spotlight)

**Comment:**

The paper focused on deep regression problems and proposed a label encoding technique which can be thought as a sibling of the famous error-correcting output codes but designed for regression problems. The main idea is well illustrated in Figure 1 at the top of page 3, where the encoder and decoder are the main objects of the proposal (and a quantizer is also needed for using the encoder/decoder which is a uniform quantizer in the paper). The idea/proposal is supported by solid theoretical arguments and convincing empirical evidences (not only the paper but also the rebuttal). While there were some concerns in the beginning, the authors have successfully clarified all the concerns and then the average score has been increased from 5.5 to 7.5. As a result, the paper is clearly above the bar of acceptance. What is more, an advantage is that the proposed method is task-agnostic and can be combined with different task-specific feature extractors borrowed from very complex regression problems (e.g., head pose estimation, facial landmark detection, age estimation, and end-to-end autonomous driving), making its significance and potential impact high. Given these facts, I think the paper might be selected as a spotlight presentation at the conference.